# Pseudo-Asynchronous Local SGD: Robust and Efficient Data-Parallel Training

**Hiroki Naganuma**[1,2,♣]**, Xinzhi Zhang**[3,♣]
*naganuma.hiroki@mila.quebec, xinzhi20@uw.edu*
[1]*Mila,* [2]*Université de Montréal,* [3]*University of Washington*

**Man-Chung Yue**[4]**, Ioannis Mitliagkas**[1,2,5]
*mcyue@hku.hkm, ioannis@mila.quebec*
[4]*The University of Hong Kong,* [5]*Canada CIFAR AI Chair*

**Philipp A. Witte**[6,♠]**, Russell J. Hewett**[7,♠,†]**, Yin Tat Lee**[3,6,♠]
*pwitte@microsoft.com, rhewett@microsoft.com, yintat@uw.edu*
[6]*Microsoft,* [7]*NVIDIA*

**Reviewed on OpenReview:** *https://openreview.net/forum?id=8VTrvS5uN7*

## Abstract

Following AI scaling trends, frontier models continue to grow in size and continue to be trained on larger datasets. Training these models requires huge investments in exascale computational resources, which has in turn driven development of distributed deep learning methods. Data parallelism is an essential approach to speed up training, but it requires frequent global communication between workers, which can bottleneck training at the largest scales. In this work, we propose a method called Pseudo-Asynchronous Local SGD (PALSGD) to improve the efficiency of data-parallel training. PALSGD is an extension of Local SGD (Stich, 2018) and DiLoCo (Douillard et al., 2023), designed to further reduce communication frequency by introducing a pseudo-synchronization mechanism. PALSGD allows the use of longer synchronization intervals compared to standard Local SGD. Despite the reduced communication frequency, the pseudo-synchronization approach ensures that model consistency is maintained, leading to performance results comparable to those achieved with more frequent synchronization. Furthermore, we provide a theoretical analysis of PALSGD, establishing its convergence and deriving its convergence rate. This analysis offers insights into the algorithm's behavior and performance guarantees. We evaluated PALSGD on image classification and language modeling tasks. Our results show that PALSGD achieves better performance in less time compared to existing methods like Distributed Data Parallel (DDP), and DiLoCo. Notably, PALSGD trains 18.4% faster than DDP on ImageNet-1K with ResNet-50, 24.4% faster than DDP on TinyStories with GPT-Neo-125M, and 21.1% faster than DDP on TinyStories with GPT-Neo–8M.

## 1 Introduction

Training neural networks has become more computationally expensive, requiring distributed deep learning techniques to handle the growing data and model sizes. Distributed training usually uses some form of data parallelism such as DDP (Li et al., 2020a) or ZeRO (Rasley et al., 2020), where a batch of training samples is further split into multiple micro batches that are assigned to different workers. These workers perform

---

♣Alphabetical order, –These authors contributed equally to this work. This work was partially done when H.Naganuma and X.Zhang were Microsoft Research interns

♠Alphabetical order

†Work done while at Microsoft

forward and backward passes on their local data shards and synchronize their model updates through operations like ALL-REDUCE. However, synchronization at every step introduces significant communication overhead, especially as the number of workers increases, because all model gradients have to be synchronized between workers (Lin et al., 2018). In addition, increasing the batch size to improve throughput can negatively impact model generalization, resulting in suboptimal performance (Keskar et al., 2016).

A common solution to reduce communication overhead is *Local SGD* (Stich, 2018), which allows workers to perform multiple local updates before synchronizing. While effective, Local SGD suffers from model divergence when synchronization intervals become too large, degrading its convergence properties (Lin et al., 2018). Recent methods such as DiLoCo (Douillard et al., 2023) and FedProx (Li et al., 2020b) introduce techniques to mitigate these issues, yet they still rely on periodic full synchronization, making them susceptible to performance bottlenecks in high-latency environments.

To address these issues, we propose *Pseudo-Asynchronous Local SGD* (PALSGD), a novel extension of the Local SGD (Stich, 2018) framework that incorporates a pseudo-asynchronous model update mechanism. Unlike standard synchronization methods that require workers to exchange gradients at fixed intervals, PALSGD allows workers to *gradually* synchronize with a local copy of the global model. Instead of waiting for an ALL-REDUCE operation, workers probabilistically mix their local parameters with an outdated version of the global model, reducing the frequency of full synchronizations. By introducing probabilistic updates, workers operate more independently between synchronization points, leading to better training efficiency. This *pseudo-synchronization* significantly alleviates communication bottlenecks while maintaining model consistency, making PALSGD particularly suitable for large-scale distributed training.

Our contributions are as follows:

- **Pseudo Synchronization**: We introduce a probabilistic pseudo-synchronization mechanism to allow workers to loosely synchronize with the global model, reducing the need for frequent full synchronization. This approach balances communication efficiency and model consistency (Section 5 and Algorithm 1).

- **Theoretical Analysis**: We provide a rigorous convergence analysis of PALSGD, demonstrating how the interplay between pseudo-synchronization probability and synchronization interval affects the overall convergence rate (Section 6).

- **Empirical Validation**: We demonstrate the effectiveness of PALSGD through experiments on ImageNet-1K (Deng et al., 2009), TinyStories (Eldan & Li, 2023), and CIFAR10 datasets. We show that it achieves superior training efficiency compared compared to existing methods like Distributed Data Parallel (DDP) and DiLoCo (Douillard et al., 2023) (Section 7, Figures 2, 3, and 4).

Our work advances the field of efficient distributed deep learning by bridging the gap between synchronous and fully decentralized framework. By introducing a flexible and efficient synchronization strategy, PALSGD provides a scalable alternative for training large-scale neural networks in real-world distributed environments.

## 2  Background

Efficient communication in distributed training is a fundamental challenge as models and datasets continue to scale. Large-scale models, such as LLaMA-3 (Dubey et al., 2024), spend a significant portion of training time—up to 20-30%—on ALL-REDUCE operations, which synchronize gradients and model states across workers. As models and datasets grow larger, this synchronization overhead becomes a major bottleneck in distributed deep learning. Local SGD (Stich, 2018), which reduces communication frequency by allowing workers to perform local updates for several steps before synchronizing, offers a partial solution. However, it was observed that Local SGD struggles when the synchronization interval $H$ exceeds about 8 steps, leading to degraded model performance and slower convergence (Lin et al., 2018). This limitation prevents Local SGD from effectively scaling to larger models, where minimizing communication overhead is essential.

Moreover, as distributed training systems scale, frequent synchronization steps reduce the system's robustness to worker slowdowns or failures. In these scenarios, a single delayed or failed worker can bottleneck the entire

training process. These issues highlight the need for communication methods that not only lower the cost and frequency of synchronization but also ensure sufficient alignment between workers' models.

## 3 Preliminaries

We consider the following stochastic optimization problem:

$$\min_{x \in \mathbb{R}^d} F(x), \quad F(x) = \mathbb{E}_{\xi \sim \mathcal{D}} f(x, \xi), \tag{1}$$

which aims to minimize the expected value of the objective function $f(x, \xi)$, where training samples $\xi$ are drawn from an underlying data distribution $\mathcal{D}$.

In distributed training, we consider a system of $K$ workers running in parallel, each initialized with the same parameter $x^{(0)}$. The training dataset is uniformly partitioned into $K$ independent data shards, $\mathcal{D}_1, \ldots, \mathcal{D}_K$, such that each worker performs local computations exclusively on its assigned subset of data. Each worker updates its local model using a general optimization rule:

$$x_k^{t+1} = \text{WORKEROPT}(x_k^t, g_k^t, \eta_t), \quad g_k^t = \nabla f(x_k^t, \xi_k^t), \tag{2}$$

where $x_k^t$ is the local model on worker $k$, $g_k^t$ is the stochastic gradient computed using the local data sample $\xi_k^t \sim \mathcal{D}_k$, and $\text{WORKEROPT}(\cdot)$ represents the local optimization algorithm (e.g., SGD or AdamW) with step size $\eta_t$.

### 3.1 Synchronization in Distributed Training

To maintain consistency across workers, distributed training requires a synchronization mechanism to aggregate updates. In our setting, this is achieved via ALL-REDUCE, a decentralized communication protocol where workers exchange and average an arbitrary state variable $S$ (which could represent model parameters, gradients, optimizer states, or other training-related variables). Specifically, after a synchronization step, all workers update their local state to the globally averaged state:

$$S^{(t)} = \frac{1}{K} \sum_{k=1}^{K} S_k^{(t)}. \tag{3}$$

Distributed Data Parallel (DDP) is a widely used distributed training algorithm. It performs an ALL-REDUCE operation on the gradient variable at each training step before applying the optimizer update. This ensures that all workers compute the same parameter updates based on the globally averaged gradient, maintaining consistency across the training process. However, frequent synchronization at every step introduces substantial communication overhead, which can significantly limit scalability as the number of workers increases.

### 3.2 The Pseudo-Asynchronous Setting

To alleviate synchronization bottlenecks, distributed training can be performed in a *pseudo-asynchronous* setting, where workers operate independently between synchronization points. In this approach, synchronization occurs only at pre-scheduled ALL-REDUCE operations, rather than at every training step. During ALL-REDUCE, all workers must reach the same iteration count before synchronization occurs, meaning that faster workers must wait for slower ones.

Between two consecutive synchronization points, workers perform local updates without waiting for each other, allowing them to progress at their own pace. This reduces total idle time and lowers communication overhead, improving computational efficiency. However, less frequent synchronization introduces the challenge of model divergence, necessitating strategies to balance communication efficiency and model consistency.

# 4 Related Work

## 4.1 Local SGD and Variants

Local SGD is a widely adopted technique for reducing communication overhead in distributed optimization. It allows workers to perform multiple local updates before synchronizing with a central server, minimizing the need for frequent gradient exchanges and improving communication efficiency. Introduced in federated learning by McMahan et al. (2017a), Local SGD has been extensively studied for its convergence properties. Stich (2018) demonstrated that it converges at the same rate as mini-batch SGD, particularly in smooth and strongly convex settings. Subsequent works, such as Haddadpour et al. (2019) and Koloskova et al. (2020), extended these results by analyzing Local SGD under more generalized frameworks, including varying network topologies, different convexity settings, and data heterogeneity. Similarly, Yu et al. (2019) provide an analysis of Local SGD, which they term Parallel Restarted SGD, further demystifying the effectiveness of model averaging in deep learning. The key insight of our algorithm is that it allows workers to mix with the "central model" more frequently without incurring extra communication overhead, leading to better alignment of local models compared to Local SGD.

Several variants of Local SGD have been proposed to further enhance scalability. Elastic Averaging SGD (EASGD) (Zhang et al., 2015) allows local models to diverge from the global model within a bounded range, improving convergence in non-convex settings. While our proposed algorithm similarly introduces slackness through proximal terms in the loss function, it further reduces communication by employing stochastic synchronization rather than stepwise synchronization. Moreover, our algorithm reduces variance between local models more effectively by ensuring that all workers start from the same global model every $H$ local steps, leading to improved consistency across workers. Cooperative SGD (Wang & Joshi, 2021) expands on Local SGD by enabling direct communication between workers, reducing reliance on a central server and improving robustness in decentralized systems. The Scaffnew algorithm (Mishchenko et al., 2022) extends Local SGD by introducing probabilistic communication updates and control variates, with communication handled through a more complex proximal optimization process. However, its reliance on exact control-variate updates makes it difficult to extend to modern optimizers like AdamW without significant modifications to the algorithm and its memory footprint. In contrast, our approach uses a simpler control mechanism, making PALSGD substantially easier to integrate into existing deep learning pipelines.

Additionally, Post-Local SGD, a combination of mini-batch SGD and Local SGD, was introduced by Lin et al. (2018) and shown to strike a better balance between communication efficiency and generalization performance in deep learning tasks (Gu et al., 2023). In our proposed method, we adopt a similar strategy to Post-Local SGD by using mini-batch SGD in the warmup phase, followed by the application of our PALSGD algorithm in the second phase. This approach allows us to leverage the fast initial convergence of mini-batch SGD before transitioning to our more communication-efficient method. SCAFFOLD (Karimireddy et al., 2020), addresses the issue of client drift by using control variates to correct the local update direction, which differs from our probabilistic mixing approach. In addition, SCAFFOLD is primarily designed to address non-IID scenarios and has been reported to perform worse than FedAvg (McMahan et al., 2017b) in IID settings. Another key related work, FedProx (Li et al., 2020b), explicitly penalizes local model divergence through a proximal term. Unlike the above-mentioned works, our method implicitly controls model alignment via probabilistic pseudo-synchronization, avoiding the need for additional hyperparameter tuning and extra forward-backward calculation. Our method also contrast with synchronous approaches like DropCompute (Giladi et al., 2023), which handles stragglers by dropping slow gradient computations rather than extending the local update period Furthermore, while our approach shares similarities with L2GD Bergou et al. (2022), we incorporate a warmup phase and an inner-outer loop optimization structure, which are particularly crucial when using AdamW for large-scale vision and language model training.

## 4.2 Negative Momentum

Momentum-based optimization methods are widely employed to accelerate the convergence of gradient-based algorithms. Recent study (Gidel et al., 2019), particularly in the context of adversarial learning such as Generative Adversarial Networks (GANs), have emphasized the role of negative momentum in improving

game dynamics. In their study, negative momentum was introduced as a stabilizing mechanism to address oscillatory behavior in adversarial settings. Their results demonstrated that alternating gradient updates with a negative momentum term achieve more efficient convergence, both theoretically and empirically, especially in bilinear games and challenging scenarios like saturating GANs.

Our proposed method, PALSGD, shares conceptual similarities with negative momentum in the context of distributed learning, despite the focus being on single-objective function optimization rather than adversarial learning. The pseudo-synchronization introduced in PALSGD can be interpreted as a form of regularization, akin to how negative momentum explicitly modifies the update direction in adversarial games. While previous study (Gidel et al., 2019) applied negative momentum to mitigate instability in adversarial games, our method regularizes model divergence in large-scale data-parallel SGD, reducing the instability often observed in such setups. Thus, negative momentum and our pseudo-asynchronous approach provide complementary insights into enhancing the stability and efficiency of gradient-based methods, albeit in distinct settings: adversarial games versus large-scale distributed learning.

### 4.3 Robust Aggregation through Decoupled Method

While Local SGD is theoretically fast-converging and communication-efficient, it faces empirical limitations in large-scale optimization tasks (Ortiz et al., 2021). One of the challenges is that simple averaging of local models, as used in Local SGD, struggles in scenarios involving adaptive optimizers like SGD momentum and AdamW, which are common in large-scale training. Recent works such as Slomo (Wang et al., 2019) and FedOpt (Reddi et al., 2020) have focused on more robust aggregation techniques by decoupling the inner optimizer for local training and the outer optimizer for model aggregation. More recent approaches such as DiLoCo (Douillard et al., 2023) and Asynchronous Local SGD (Liu et al., 2024) have validated the effectiveness of using AdamW for local updates and Nesterov momentum for outer optimization in large-scale language modeling tasks, offering improved performance and robustness. Our proposed algorithm intergrates the decoupled method from the DiLoCo framework with the pseudo-synchronization process. We showed that our method significantly outperforms DiLoCo on image classification and language modeling tasks.

### 4.4 Asynchronous and Pseudo-Asynchronous Methods

Asynchronous and pseudo-asynchronous methods have been developed to address inefficiencies in synchronous training, particularly the "straggler effect", where faster workers are forced to wait for slower ones. This issue has been widely observed in synchronous distributed settings (Koh et al., 2006; Lian et al., 2015; 2018; Dean et al., 2012). Dean et al. (2012) introduced one of the earliest asynchronous frameworks, enabling each worker to update the global model independently, which significantly improved computational utilization. However, this approach introduced the challenge of stale gradients, where outdated updates from slower workers are applied to newer models, hindering convergence. Methods like Asynchronous SGD with Delay Compensation (Zheng et al., 2017) addressed this issue by approximating fresher gradients. Other approaches such as Polyak Averaging (Xie et al., 2019) proposed downweighting stale updates to improve robustness. More recently, Asynchronous Local SGD (Liu et al., 2024) introduced a decoupled method, demonstrating that the strategic use of momentum can alleviate many of the challenges posed by staleness. In federated learning, methods like Moshpit SGD (Ryabinin et al., 2021) and TimelyFL (Zhang et al., 2023) have explored asynchronous approaches to better manage unreliable or heterogeneous devices in large-scale distributed systems. Tyurin & Richtárik (2024) proposes a time complexity framework for parallel stochastic optimization under a fixed computation model.

Recent works have also explored fully asynchronous and decentralized settings to further mitigate synchronization bottlenecks. Methods like Shadowheart SGD (Tyurin et al., 2024) and SWIFT (Bornstein et al., 2022) leverage parameter-server architectures to enable wait-free communication, while others have explored techniques like model fragmentation to boost asynchronous learning (Biswas et al., 2025). These approaches differ from our semi-synchronous framework, which maintains periodic global synchronization points to avoid issues like unbounded gradient staleness.

# 5 Proposed Method: PALSGD

---

**Algorithm 1:** Pseudo-Asynchronous Local SGD with Decoupled Optimizers

---

**Data:** $x^{(0)}$ (initial model), $K > 0$ (number of workers), $p \in (0,1)$ (probability of mixing step), $\eta_t > 0$ (mixing rate), $H > 0$ (sync interval), optimizers INNEROPT and OUTEROPT, $\alpha_t$ (learning rate for INNEROPT)

**for** *worker* $k = 1, \cdots, K$ **do**

    $x_k^{(0)} \leftarrow x^{(0)}$;

    **for** $t = 0, \cdots, T - 1$ **do**

        $b \sim U[0,1]$;

        **if** $b \leq p$ **then**

            $x_k^{(t)} \leftarrow x_k^{(t)} - \frac{\alpha_t \eta_t}{p} \cdot (x_k^{(t)} - x^{(t)})$;    `pseudo-synchronization step`

        **else**

            Sample data $\xi \sim \mathcal{D}_k$;

            $g_k^{(t)} \leftarrow \nabla f(x_k^{(t)}, \xi)$;

            $x_k^{(t+1)} \leftarrow$ INNEROPT$(x_k^{(t)}, g_k^{(t)}, \frac{\alpha_t}{1-p})$;    `gradient step`

        **end**

        **if** $(t+1) \mod H = 0$ **then**

            $\Delta^{(t)} \leftarrow$ ALL-REDUCE$(x^{(t-1)} - x_k^{(t)})$;    `aggregate outer gradient`

            $x^{(t+1)} \leftarrow$ OUTEROPT$(x^{(t)}, \Delta^{(t)})$;    `update global model`

        **else**

            $x^{(t+1)} \leftarrow x^{(t)}$

        **end**

    **end**

**end**

---

The primary challenge in extending the synchronization interval $H$ in Local SGD is the problem of model divergence, where worker models drift from the global optimum and degrade convergence. To address this, we propose Pseudo-Asynchronous Local SGD (PALSGD), a novel extension of the Local SGD framework. PALSGD introduces a pseudo-synchronous step that acts as a lightweight, communication-free regularizer to explicitly counteract this divergence. As detailed in Algorithm 1, this step allows workers to probabilistically align with a locally stored global model copy, enabling the use of longer synchronization intervals without sacrificing model consistency.

## 5.1 Probablistic Regularization via Pseudo-Synchronization

In our distributed setting, each worker maintains two key model states: a *local model* $x_k^{(t)}$, which is updated using data from its assigned shard, and a *global model copy* $x^{(t)}$, which stores the last synchronized state and remains unchanged between full synchronization steps. Training progresses over multiple iterations, where each worker performs local updates independently for $H$ steps before a full synchronization occurs.

At each local step, the worker follows one of two paths based on a probability $p$. With probability $1 - p$, it performs a standard gradient-based update using its local data and the optimizer INNEROPT:

$$x_k^{(t+1)} = \text{INNEROPT}(x_k^{(t)}, g_k^{(t)}, \frac{\alpha_t}{1-p}), \tag{4}$$

where $g_k^{(t)} = \nabla f(x_k^{(t)}, \xi)$ is the stochastic gradient computed from the worker's local data shard.

Alternatively, with probability $p$, the worker performs a pseudo-synchronization update:

$$x_k^{(t+1)} \leftarrow x_k^{(t)} - \frac{\alpha_t \eta_t}{p}(x_k^{(t)} - x^{(t)}). \tag{5}$$

This update is the core of PALSGD and serves as a principled mechanism to control model divergence. Its form can be justified in several ways:

- **As a Proximal Regularizer:** The update in Equation (2) can be interpreted as a single gradient step on a quadratic penalty term, $\frac{\eta_t}{2}\|x_k - x^{(t)}\|^2$. This term explicitly regularizes the local model by pulling it towards the last-known global state $x^{(t)}$. It penalizes deviation from this shared reference point, thereby constraining the local optimization paths and preventing workers from drifting too far apart.

- **As an Exponential Moving Average (EMA):** The update can be rewritten as a mixing step: $x_k^{(t+1)} \leftarrow (1 - \beta)x_k^{(t)} + \beta x^{(t)}$, where the mixing coefficient is $\beta = \frac{\alpha_t \eta_t}{p}$. This shows that the worker's parameters are an exponential moving average of their own trajectory and the global trajectory. This form is well-known for stabilizing updates by dampening oscillations and reducing the variance of the parameter updates.

By probabilistically applying this step, PALSGD ensures that local models remain sufficiently aligned, acting as a lightweight mechanism to reduce model divergence without incurring any communication cost. We refer to this as a *pseudo-synchronization* step to distinguish it from a true synchronization event like ALL-REDUCE. In a true synchronization, workers exchange information and compute a global average, which requires all workers to pause and communicate. In contrast, a pseudo-synchronization step is a purely local operation. Each worker moves its local model towards its own stored copy of the global model, which may be stale by up to $(H-1)$ steps. This step involves no communication with other workers, thus avoiding network latency and idle time for faster workers.

## 5.2 Reducing the Frequency of Full Synchronization

By employing the probabilistic regularization described above to maintain model consistency, PALSGD can safely reduce the frequency of full, communication-heavy synchronizations. Unlike fully synchronous methods where workers communicate at every iteration, PALSGD leverages pseudo-synchronization to partially align workers' models, reducing the need for frequent global updates. This approach mitigates the problem of idle time, where faster workers must wait for slower ones, and improves scalability, especially in large distributed systems.

A full global synchronization occurs only every $H$ iterations via an ALLREDUCE operation, which aggregates the model differences across all $K$ workers:

$$\Delta^{(t)} = \text{ALLREDUCE}(x^{(t-1)} - x_k^{(t)}). \tag{6}$$

The aggregated difference $\Delta^{(t)}$ is then used to update the global model with an outer optimization step:

$$x^{(t+1)} \leftarrow \text{OUTEROPT}(x^{(t)}, \Delta^{(t)}). \tag{7}$$

By performing this costly operation less frequently, PALSGD significantly reduces communication overhead while the pseudo-synchronization mechanism preserves model consistency between these updates.

## 5.3 Practical Enhancements and Implementation Details

To further improve empirical performance and ensure stability, PALSGD incorporates several practical techniques with minimal overhead. First, similar to Post-Local SGD (Lin et al., 2018), we initialize training with a DDP warm-up phase to address potential instability when gradients are large and noisy. Following this, we employ a decoupled optimizer strategy, as validated in the DiLoCo framework (Douillard et al., 2023). We use AdamW (Kingma, 2014) as the INNEROPT for rapid local progress and Nesterov momentum (Sutskever et al., 2013) as the OUTEROPT to robustly aggregate updates.

From an implementation standpoint, these enhancements are lightweight:

- **Memory Requirements:** Each worker stores the local model $x_k^{(t)}$, the global model copy $x^{(t)}$, and the outer optimizer state. This overhead, equivalent to one extra model and its optimizer state, is well within the capacity of modern GPUs.

- **Communication Requirements:** The pseudo-synchronization step is a purely local computation and adds no communication overhead. Communication occurs only during the ALL-REDUCE operation every $H$ steps. The primary efficiency gain of PALSGD stems from its ability to use a larger $H$ effectively, thereby reducing the total number of these expensive communication events.

Together, these modifications ensure that PALSGD not only reduces communication costs but also achieves faster convergence and better model performance across various deep learning tasks.

## 6 Theoretical Results

Building on this framework, we showed the following theoretical convergence bound for a simplified version of our algorithm, where the inner optimizer is standard SGD and the outer model is updated by taking the average across inner models. We include the proof in Appendix A.

We make the following assumptions on the objective function $F$ for our theoretical analysis:

**Assumption 1** ($L$-Smoothness). *There exists a constant $L > 0$ such that for each $\xi$ in the support of $\mathcal{D}$, and for each $x, y \in \mathbb{R}^d$,*

$$\|\nabla f(x, \xi) - \nabla f(y, \xi)\| \leq L\|x - y\|.$$

**Assumption 2** ($\mu$-Strongly Convex). *There exists a constant $\mu > 0$ such that for each $\xi$ in the support of $\mathcal{D}$, $f(x, \xi)$ is $\mu$-strongly convex. Moreover, write $x^* = \arg\min_{x \in \mathbb{R}^d} F(x)$ as the global minimal solution.*

**Assumption 3** (Identical Data Distributions among Workers). *Let $\mathcal{D}_1, \mathcal{D}_2, \ldots, \mathcal{D}_K$ be the data distributions for $K$ workers. Assume that these distributions are identical and independent, denoted by $\mathcal{D}_1 = \mathcal{D}_2 = \cdots = \mathcal{D}_K = \mathcal{D}$.*

**Assumption 4** (Bounded Variance at Optimal). *There exists $\sigma \geq 0$ such that*

$$\mathbb{E}_{\xi \sim \mathcal{D}}[\|\nabla f(x^*, \xi)\|^2] \leq \sigma^2.$$

**Theorem 1** (Convergence of PALSGD, Informal). *Let $x^{(0)}, \cdots, x^{(T-1)}$ be the sequence generated by Algorithm 1 with INNEROPT as SGD and OUTEROPT as SGD with step size 1. Under Assumptions 1, 2, 3, and 4, let $\kappa = \frac{L}{\mu}$. For any $0 < p \leq \frac{1}{2}$, and for any $T > 0$, there exists a sequence of inner step size $\{\alpha_t\}_{t=0}^{T-1}$, a sequence of mixing rate $\{\eta_t\}_{t=0}^{T-1}$, and a weight sequence $\{w_t\}_{t=0}^{T-1}$ such that for $\hat{x}_T = \frac{1}{Z_T} \sum_{t=0}^{T-1} w_t x^{(t)}$ where $Z_T = \sum_{t=0}^{T-1} w_t$, and ignorining the logrithmic and exponentially decaying terms, we have*

$$\mathbb{E}[F(\hat{x}_T)] - F(x^*) \leq \tilde{O}\left(\frac{\sigma^2}{\mu KT} + \frac{\kappa H^2 \sigma^2}{\mu T^2}\right). \tag{8}$$

The convergence bound in Theorem 1 contains two parts. The first term, $\frac{\sigma^2}{\mu KT}$, reflects a linear speedup with respect to the number of workers $K$ and is independent of the hyperparameters for the pseudo-synchronization steps. When $T$ is sufficiently large, this term dominates and matches the Cramer-Rao bound for estimating a single random variable. The second term arises from the stochastic optimization process. As $K$ increases and communication becomes frequent (i.e., $H$ is small constant), we can simplify the error bound to $\frac{\kappa \sigma^2}{\mu T^2}$, which matches the lower bound for stochastic gradient descent in the strongly convex case (Koloskova et al., 2020).

## 7 Experiments

### 7.1 Experimental Setup

We performed experiments on five different workloads to assess the performance of PALSGD in a distributed training setup. For image classification, we used CIFAR-10 (Krizhevsky et al., 2009) with small CNN and

VGG16(Simonyan & Zisserman, 2014), ImageNet-1K (Deng et al., 2009) with ResNet-50 (He et al., 2016). For language modeling, we employed TinyStories (Eldan & Li, 2023), using GPT-Neo[1] with 8 million and 125 million parameters.

To ensure a fair comparison, we evaluated PALSGD alongside three widely used distributed training baselines: Distributed Data Parallel (DDP), Local SGD, and DiLoCo (Douillard et al., 2023). In the CIFAR-10 experiments, we varied the number of workers from 4 to 64. For ImageNet-1K experiments, we trained the models with $K = 4$ workers, while for the TinyStories experiments, we trained the models using 4 to 8 workers.

Following the standard approach for post-Local SGD method (Lin et al., 2018), all three algorithms — Local SGD, DiLoCo, and PALSGD - used DDP as warmup during the initial training steps. This warmup phase stabilizes training and improves model initialization before transitioning to a more communication-efficient distributed method. The training hyperparameters of all algorithms, including learning rates and $\eta$, were tuned independently for each choice of $K$ (number of workers) and $H$ (synchronization interval). This tuning process ensures that each algorithm is evaluated under its best-performing configuration for a given distributed training setting. Further details including the ablation studies are provided in Appendix B and E.

## 7.2 Preliminary Experiments: CIFAR10 on Small CNN

We begin by presenting small-scale results, including comparisons with Local SGD (Ortiz et al., 2021). DiLoCo (Douillard et al., 2023) can be interpreted as equivalent to Local SGD, or to FedAvg (McMahan et al., 2017b) in the IID case, when the outer optimizer is set to SGD. As shown in Figure 6 of the Douillard et al. (2023), DiLoCo outperforms these methods. We conducted preliminary experiments to examine how accuracy changes as the number of workers increases for DDP, Local SGD, DiLoCo, and PALSGD. The results are shown in Figure 1. The synchronization interval was fixed at 32.

The results indicate that LocalSGD degrades substantially as the number of workers increases, compared to the other three methods. Our result consistents with observations in Lin et al. (2018). DiLoCo and PALSGD show similar performance when worker counts or synchronization intervals are small. In contrast, PALSGD maintains better performance when synchronization intervals become large. This occurs because DiLoCo permits extended intervals without ensuring model consistency. PALSGD addresses this issue by imposing pseudo-synchronization constraints.

Based on these findings, we do not include Local SGD in large-scale experiments. Instead, we use DDP as the baseline and DiLoCo as the main comparison method for evaluating PALSGD. Methods such as SCAFFOLD(Karimireddy et al., 2020) are excluded, since prior work has shown that they underperform LocalSGD in the IID setting with reasonable epochs budget (see Table 3 in Karimireddy et al. (2020)).

The following subsections report results from more practical experiments on ImageNet and TinyStories.

## 7.3 Image Classification Tasks: ImageNet-1K on ResNet 50

In the ImageNet-1K experiments using ResNet-50, we measured the time required for different training algorithms to reach a specific validation accuracy. All three methods—PALSGD, DiLoCo, and DDP—achieved the same target top-1 accuracy of 75% (which is the standard thresholds for image classification tasks Lin et al. (2018); Mattson et al. (2020)), but PALSGD demonstrated an 18.4% speedup in terms of training steps compared to DDP and a 6% speedup over DiLoCo (See Figure 2 for details). This highlights PALSGD's ability to improve training efficiency while maintaining competitive accuracy.

Unlike previous findings on Local SGD Ortiz et al. (2021), PALSGD's final validation accuracy was not lower than DDP, as all methods converged to the same 75% target. This result contrasts with prior concerns that PALSGD might suffer from slight accuracy degradation in vision tasks. Given that ImageNet-1K was not originally evaluated in the DiLoCo paper (Douillard et al., 2023), our experiments provide a more comprehensive comparison across different domains. To further support our results on image classification

---

[1] https://huggingface.co/docs/transformers/en/model_doc/gpt_neo

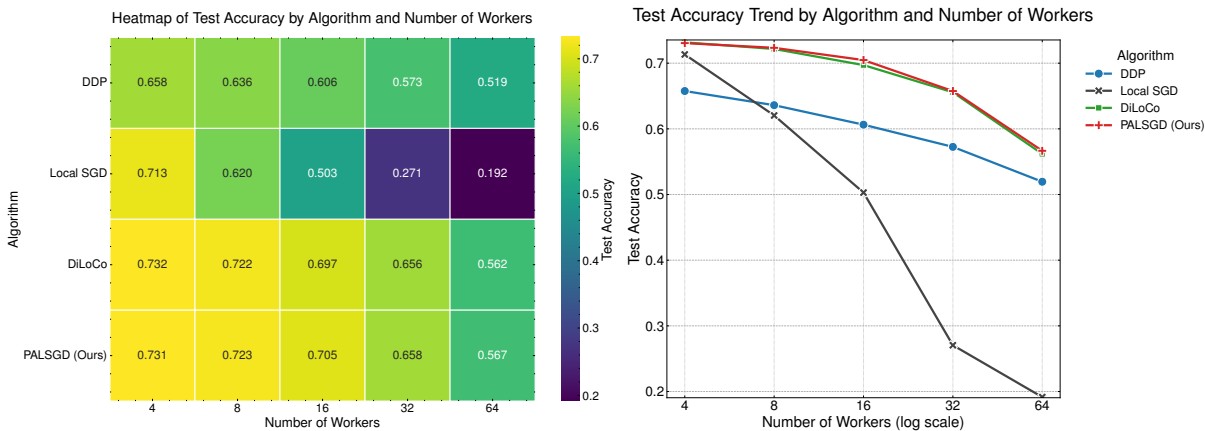

Figure 1: Comparison of number of workers across algorithm / simulation experiments with CIFAR10 dataset

workloads, we also evaluated the effectiveness of our method on a workload involving CIFAR-10 training using VGG-16 (Simonyan & Zisserman, 2014). As a result, PALSGD demonstrated superior performance compared to DiLoCo and DDP. Details are provided in Appendix C.4.

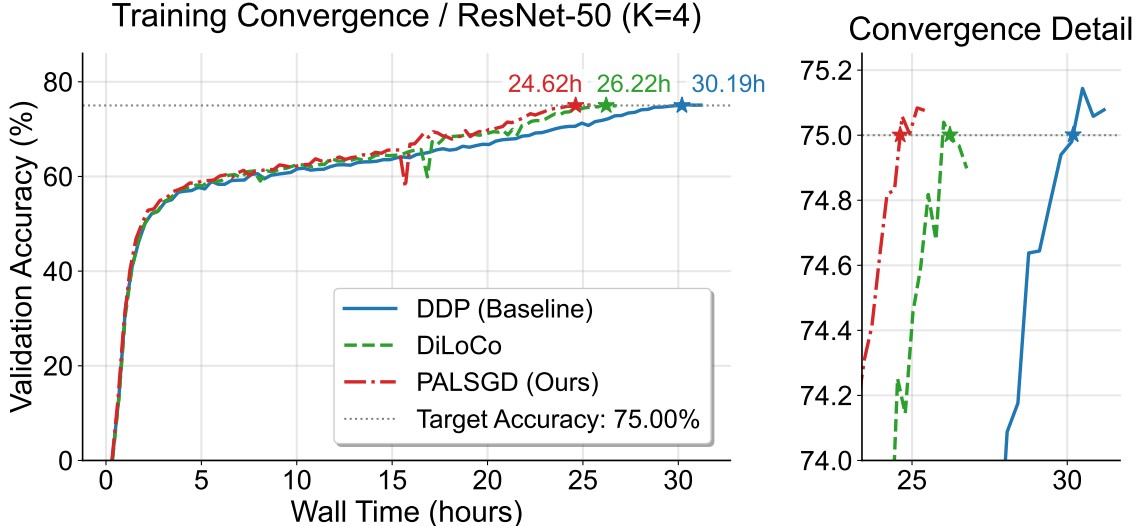

Figure 2: ImageNet-1K on ResNet-50 Experiments ($K = 4$ / $H = 64$): PALSGD demonstrates faster training compared to DDP and DiLoCo, while achieving the same target validation accuracy. The results regarding training accuracy are presented in Figure C.1 in Appendix C.1.

### 7.4 Language Modeling Tasks: TinyStories on GPT-Neo

In the TinyStories experiments on GPT-Neo with $K = 8$, PALSGD demonstrated significant reductions in communication overhead by minimizing the frequency of synchronization steps. Specifically, PALSGD reduced the total number of synchronization steps by 93.75% compared to DDP[2]. As a result, in the GPT-Neo-125M experiment, PALSGD was 24.4% faster than DDP while achieving the same target validation loss, as shown in Figure 3. Similarly, in the GPT-Neo-8M experiment, PALSGD reduced the total training time by 21.1% while still reaching the target validation loss, as depicted in Figure 4. In contrast, DiLoCo converged more slowly and failed to reach the target loss within the same training timeframe. Comparable results were also observed for the case of $K = 4$, with details provided in Appendix C.2.1. These findings further

---

[2]This is the theoretical value when the communication frequency is reduced to one-sixteenth.

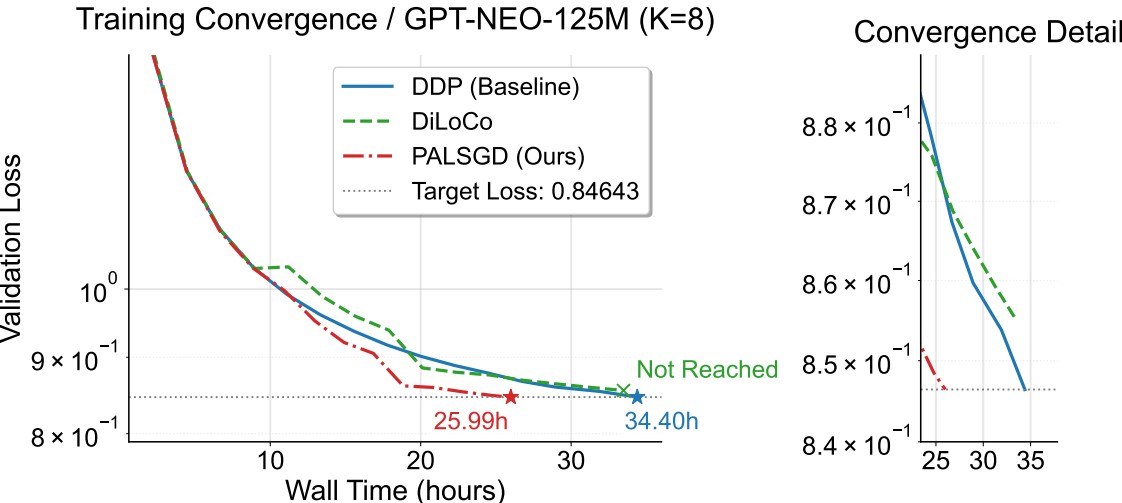

Figure 3: GPT-Neo-125M Experiments (K=8 / H=16): Training time comparison across distributed algorithm to achieve target validation loss. While PALSGD achieves fastest and lowest loss, DDP is slowest and DiLoCo did not achieve target loss. The results regarding training accuracy are presented in Figure C.4 in Appendix C.3.

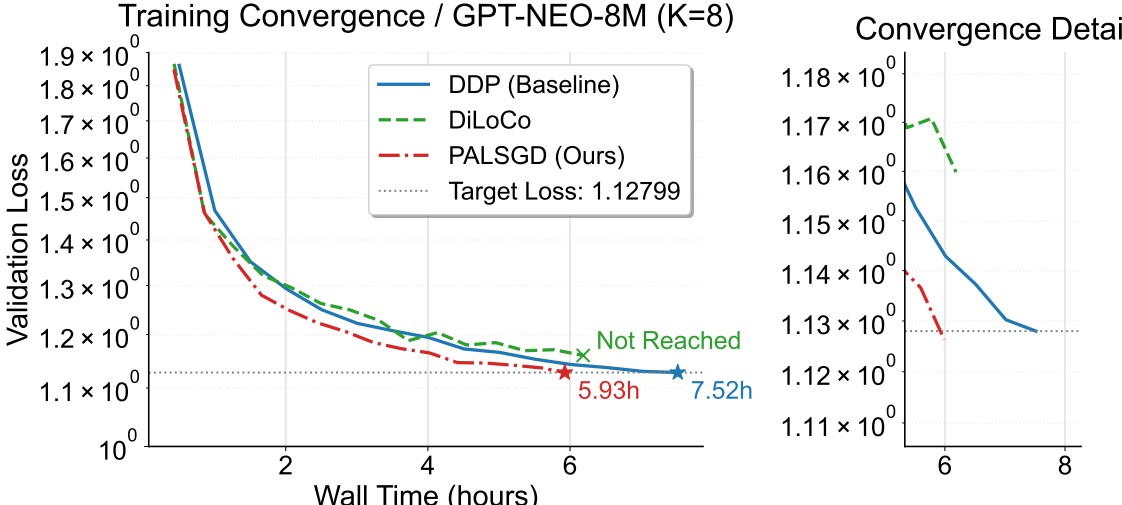

Figure 4: GPT-Neo-8M Experiments (K=8 / H=16): Training time comparison across distributed algorithm to achieve target validation loss. PALSGD achieves fastest and lowest loss, DDP is slowest and DiLoCo did not achieve target loss. The results regarding training accuracy are presented in Figure C.3 in Appendix C.2.2.

confirm PALSGD's effectiveness in reducing communication cost without compromising model quality or convergence speed.

Our experiments demonstrate that PALSGD effectively balances communication efficiency and model performance. The probabilistic pseudo-synchronization mechanism allows workers to update their local models independently, leading to faster convergence and reduced communication overhead. Compared to existing methods, PALSGD achieves significant improvements in both training speed and model loss, making it a compelling choice for scalable distributed training in modern language model workloads.

In addition, we performed a series of ablation studies on the hyperparameters of GPT-Neo-8M experiment, and we present the result of synchronization interval $H$ here. Ablation studies on other hyperparame-

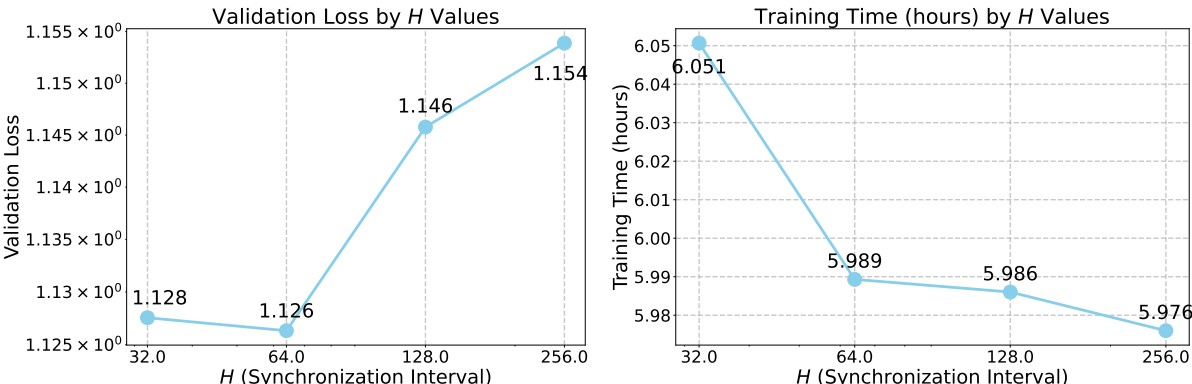

Figure 5: Comparison of $H$, GPT-Neo-8M Experiments ($K = 8, \eta = 16$). (Left) Validation Loss Sensitivity Analysis of $H$, (Right) $H$ Effects for Training Time.

ters—including the optimizer, random seeds, $\eta$, and $p$—are provided in Appendix E. Overall, across all workloads we observed a consistent trend: values of $p \geq 0.25$ led to faster training but degraded performance, while in practical applications such as ImageNet-1K and TinyStories, values around 0.05 yielded better results.

For synchronization interval, a smaller $H$ leads to longer training times due to more frequent communication during explicit synchronization. Conversely, a larger $H$ reduces the number of ALL-REDUCE operations, thereby shortening the time needed to complete an epoch. To assess the effect of $H$, we conducted an ablation study using GPT-Neo-8M with $K = 8$ workers. The results, shown in Figure 5, illustrate how different values of $H$ impact validation loss (left) and training time (right).

Because the outer learning rate was tuned for $H = 64$, we observed improved loss and accuracy at this value compared to smaller $H$. However, increasing $H$ beyond 64 (e.g., $H \geq 128$) led to degraded validation loss, likely due to excessive local updates causing model drift before synchronization. Meanwhile, training time consistently decreased with larger $H$, indicating that less frequent synchronization effectively reduces communication overhead. These results highlight a key trade-off between training efficiency and model performance. They suggest that moderate values of $H$ can offer the best balance for large-scale distributed training.

## 8 Discussion and Conclusion

We introduced Pseudo-Asynchronous Local SGD (PALSGD), which reduces communication overhead in large-scale distributed learning through probabilistic pseudo-synchronization. Extending Local SGD, PALSGD decreases the frequency of ALL-REDUCE operations, allowing for extended local updates. This method is particularly effective in high-latency environments, such as intercontinental data centers, where it enables more efficient, scalable training.

**Summary of Contributions**

Our key contributions are as follows:

- We introduced PALSGD, a novel extension of Local SGD that incorporates probabilistic pseudo-synchronization, significantly reducing the cost of synchronization without sacrificing model performance (Section 5 and Algorithm 1).

- We provided theoretical convergence bounds for a simplified version of PALSGD (Section 6).

- We empirically validated PALSGD on image classification and language modeling tasks, demonstrating its effectiveness in reducing training time and improving model performance compared to baseline methods such as DDP, Local SGD and DiLoCo (Section 7, Figures 2, 4, and 3).

**Limitations and Future Work**

The limitations of our work include several factors.

First, our current approach assumes homogeneous hardware and network configuration across all workers. Future research could explore adaptive methods to address heterogeneous environments, where worker speeds or network latencies vary.

Second, our theoretical analysis was simplified, focusing on PALSGD with SGD as the inner and outer optimizer and assuming strongly convex functions, whereas training deep models is inherently non-convex. Extending this theoretical framework to more complex optimizers like Adam or other adaptive methods may offer deeper insights into the algorithm's performance.

Third, we do not include asynchronous baselines in this work. These methods have been studied extensively in Liu et al. (2024); Hadjis et al. (2016), and including them here would not only be redundant but could also obscure the focus of our study on semi-synchronous approaches. For this reason, we explicitly state their exclusion as a limitation here.

Finally, while PALSGD enhances communication efficiency, future studies could investigate further reducing synchronization costs, for example, by employing gradient compression techniques or decentralized communication patterns.

# Broader Impact Statement

Our work improves the efficiency of distributed deep learning by reducing communication overhead, making large-scale model training more accessible and cost-effective. These advancements can benefit various applications, including healthcare, scientific research, and industry-scale AI deployments. As our method primarily focuses on optimization and scalability, we do not foresee significant societal risks or ethical concerns.

# Acknowledgement

Our deepest gratitude goes out to the anonymous reviewers and the Action Editor, whose invaluable insights substantially enhanced the quality of this manuscript. We also acknowledge that this research was enabled in part by computing resources, software, and technical assistance provided by Mila and the Digital Research Alliance of Canada. Man-Chung Yue is supported in part by Hong Kong Research Grants Council under the GRF project 15304422.

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

## Appendix Table of Contents

# A   Missing Proofs

In this section, we prove the convergence theorem for PALSGD in the strongly convex case, where both the inner and outer optimizers are gradient descent.

## A.1   Main Result

We restate the main theorem as below:

**Theorem 2** (Convergence of PALSGD). *Let $x^{(0)}, \cdots, x^{(T-1)}$ be the sequence generated by Algorithm 1 with INNEROPT as SGD and OUTEROPT as SGD with step size 1. Under Assumptions 1, 2, 3, and 4, for any $0 < p \le \frac{1}{2}$, let $\alpha_t = \alpha$, $\eta_t = \eta = \frac{p}{2H\alpha}$, and $w_t = (1 - \mu\alpha)^{-(t+1)}$ where*

$$\alpha = \min\left( \frac{p}{48LH}, \frac{\ln(\mu^2 d_0 T^2 K/\sigma^2)}{\mu T} \right)$$

*For any $T > 0$, let $\hat{x}_T = \frac{1}{Z_T} \sum_{t=0}^{T-1} w_t x^{(t)}$ where $Z_T = \sum_{t=0}^{T-1} w_t$, we have*

$$\mathbb{E}[F(\hat{x}_T)] - F(x^*) \le \tilde{O}\left( \frac{LHR_0^2}{p} \cdot \exp(-\frac{pT}{\kappa H}) + \frac{\sigma^2}{\mu K T} + \frac{\kappa H^2 \sigma^2}{\mu T^2} \right).$$

*where $R_0 = \|x^{(0)} - x^*\|^2$ and $\kappa = \frac{L}{\mu}$.*

The roadmap for the remainder of this section is as follows: Section A.2 introduces the basic definitions used in the proof. Section A.3 provides a proof sketch and presents the main technical lemmas. Section A.4 contains the full proof of Theorem 2. Finally, Section A.5 proves the technical lemmas stated in Section A.3.

## A.2   Basic Definitions

For any $k \in [K], t = 0, \cdots, T-1$, let $b_k^{(t)}$ denote a Bernoulli random variable with parameter $p$ (i.e., $b_k^{(t)} = 1$ with probability $p$ and $b_k^{(t)} = 0$ with probability $1 - p$). Let

$$g_k^{(t)} = \frac{1 - b_k^{(t)}}{1 - p} \nabla f(x_k^{(t)}, \xi_k^{(t)}) + \frac{\eta_t b_k^{(t)}}{pK}(x_k^{(t)} - x^{(t)}).$$

The for any $t$ such that $(t + 1) \mod H \ne 0$, we can rewrite the inner step as

$$x_k^{(t+1)} = x_k^{(t)} - \alpha_t\, g_k^{(t)}.$$

Let

$$g^{(t)} = \frac{1}{K} \sum_{k \in [K]} g_k^{(t)}. \tag{9}$$

Let $\bar{x}^{(t)} = \frac{1}{K} \sum_{k \in [K]} x_k^{(t)}$ as the current mean of the client servers at step $t$. Then we have

$$\bar{x}^{(t+1)} = \bar{x}^{(t)} - \alpha_t\, g^{(t)}.$$

Let $\xi_k^{(t)}$ denote the data sampled by the $k$-th server at step $t$. Let $\mathcal{F}_t = \{\xi_k^{(s)}\}_{s=0,\cdots,t-1,k\in[K]} \cup \{b_k^{(s)}\}_{s=0,\cdots,t-1,k\in[K]}$ for $t \ge 1$ and $\mathcal{F}_0 = \emptyset$. Define $\bar{g}^{(t)}$ as the expectation of $g^{(t)}$ over the randomness at step $t$, i.e.

$$\bar{g}^{(t)} = \mathbb{E}[g^{(t)} \mid \mathcal{F}_t] = \frac{1}{K} \sum_{k \in [K]} \left( \nabla F(x_k^{(t)}) + \eta_t(x_k^{(t)} - x^{(t)}) \right). \tag{10}$$

For any $t \ge 0$, let $t^- \le t$ be the largest integral multiples of $H$ that is at most $t$, i.e. $t^-$ is the last iteration at or before $t$ such that $x^{(t)}$ is updated. Similarly, let $t^+ \ge t$ be smallest integral multiples of $H$ that is at least $t$, i.e. the next round at or after $t$ such that $x^{(t)}$ is updated.

### A.3 Main Technical Lemmas

We now sketch the proof of Proof of Theorem 2. Our analysis employs the framework of Local SGD (Stich, 2018). The main technical lemmas are as follows:

**Lemma 1.** *Under Assumption 1 with L and Assumption 2 with $\mu \geq 0$, for any $t \geq 0$ with $\alpha_t \leq \frac{1}{4L}$, it holds that*

$$\mathbb{E}\left[\|\bar{x}^{(t+1)} - x^*\|^2\right] \leq (1 - \mu\alpha_t)\mathbb{E}\left[\|\bar{x}^{(t)} - x^*\|^2\right] + \alpha_t^2 \mathbb{E}\left[\|g^{(t)} - \bar{g}^{(t)}\|^2\right]$$
$$- \frac{\alpha_t}{2}\mathbb{E}\left[F(\bar{x}^{(t)}) - F(x^*)\right] + \frac{2\alpha_t L}{K}\sum_{k \in [K]}\mathbb{E}\left[\|x_k^{(t)} - \bar{x}^{(t)}\|^2\right].$$

Lemma 1 can be proved almost verbatim to (Stich, 2018, Lemma 3.1).

**Lemma 2.** *Under Assumption 1 with $L \geq 0$, Assumption 3 and Assumption 4. Suppose that $p \leq \frac{1}{2}$, $12L^2 \leq \eta_t^2/p$, and $\mathbb{E}_{\xi \sim \mathcal{D}}[\|\nabla f(x^*, \xi)\|^2] \leq \sigma^2$. Let $g^{(t)}$ and $\bar{g}^{(t)}$ be defined as equation 9 and equation 10 respectively. Then for any $t \geq 0$, it holds that*

$$\mathbb{E}\left[\|g^{(t)} - \bar{g}^{(t)}\|^2\right] \leq \frac{3\eta_t^2(1-p)}{p} \cdot \frac{1}{K^2}\sum_{k=1}^{K}\mathbb{E}[\|x_k^{(t)} - x^{(t)}\|^2] + \frac{24L}{K^2}\sum_{k=1}^{K}\mathbb{E}[F(\bar{x}^{(t)}) - F(x^*)] + \frac{12\sigma^2}{K}.$$

The proof of Lemma 2 can be found in Section A.5.1.

We then show the following lemma that upper-bounds $\frac{1}{K}\sum_{k \in [K]}\mathbb{E}[\|x_k^{(t)} - x^{(t)}\|^2]$ by a recursion:

**Lemma 3.** *Suppose $0 < p \leq \frac{1}{2}$ and the sequences $\{\alpha_t\}_{t \geq 0}, \{\eta_t\}_{t \geq 0}$ satisfy (1) $\alpha_t\eta_t = \frac{p}{2H}$ and (2) $\alpha_t \leq \frac{p}{6LH}$. Suppose that $\mathbb{E}_{\xi \sim \mathcal{D}}[\|\nabla f(x^*, \xi)\|^2] \leq \sigma^2$. Then for any $t \geq 0$, if $(t+1) \mod H = 0$ we have $\Xi_t = 0$, if $(t+1) \mod H \neq 0$ we have*

$$\frac{1}{K}\sum_{k \in [K]}\mathbb{E}[\|x_k^{(t)} - x^{(t)}\|^2]$$

$$\leq 12LH \cdot \sum_{s=t^-}^{t-1}\alpha_s^2\frac{1}{K}\sum_{k \in [K]}\mathbb{E}[F(x^{(s)}) - F(x^*)] + 6H^2\alpha_{t^-}^2\sigma^2 + \frac{p^2}{2H} \cdot \sum_{s=t^-}^{t-1}\frac{1}{K}\sum_{k \in [K]}\mathbb{E}[\|x_k^{(s)} - x^{(t^-)}\|^2].$$

The proof of Lemma 3 can be found in Section A.5.2.

We further simplify the recurrence of $\frac{1}{K}\sum_{k \in [K]}\mathbb{E}[\|x_k^{(t)} - x^{(t)}\|^2]$ by taking weighted sum from $t = 0$ to $T$.

**Lemma 4.** *Suppose the sequences $\{\Xi_t\}_{t \geq 0}, \{e_t\}_{t \geq 0}$ satisfy (1) for all $(t+1) \mod H = 0$ $\Xi_t = 0$, and (2) for all $(t+1) \mod H \neq 0$,*

$$\Xi_t \leq \frac{p}{2H} \cdot \sum_{s=t^-}^{t-1}\Xi_s + 6H\alpha_{t^-}^2\sum_{s=t^-}^{t-1}\sigma^2 + 12LH \cdot \sum_{s=t^-}^{t-1}\alpha_s^2 e_s. \tag{11}$$

*Suppose that for all $t \geq 0$, $\alpha_t = \alpha \leq \frac{p}{6LH}$ and $w_t \leq w_{t+1} \leq (1 + \frac{p}{H})w_t$. Then for all $T > 0$ we have*

$$\sum_{s=0}^{T}w_s\Xi_s \leq 9\alpha^2 H^2\sum_{s=0}^{T}w_s\sigma^2 + \frac{p^2}{48L}\sum_{s=0}^{T}w_s e_s.$$

The proof of Lemma 4 can be found in Section A.5.3.

### A.4 Proof of Theorem 2

We are now able to show Theorem 2 using the previous lemmas.

*Proof.* Combining with Lemma 1 and Lemma 2, we have

$$
\begin{aligned}
\mathbb{E}\left[\|\bar{x}^{(t+1)} - x^*\|^2\right] &\leq (1 - \mu\alpha_t)\,\mathbb{E}\left[\|\bar{x}^{(t)} - x^*\|^2\right] + \left(\frac{24L\alpha_t^2}{K^2} - \frac{\alpha_t}{2}\right)\mathbb{E}\left[F(\bar{x}^{(t)}) - F(x^*)\right] \\
&\quad + \left(\frac{2\alpha_t L}{K} + \frac{3\eta_t^2\alpha_t^2(1-p)}{pK^2}\right)\sum_{k\in[K]}\mathbb{E}\left[\|x_k^{(t)} - \bar{x}^{(t)}\|^2\right] + \frac{12\alpha_t^2\sigma^2}{K} \\
&\leq (1 - \mu\alpha_t)\,\mathbb{E}\left[\|\bar{x}^{(t)} - x^*\|^2\right] + \left(\frac{24L\alpha_t^2}{K^2} - \frac{\alpha_t}{2}\right)\mathbb{E}\left[F(\bar{x}^{(t)}) - F(x^*)\right] \\
&\quad + \left(\frac{8\alpha_t L}{K} + \frac{12\eta_t^2\alpha_t^2(1-p)}{pK^2}\right)\sum_{k\in[K]}\mathbb{E}[\|x_k^{(t)} - x^{(t)}\|^2] + \frac{12\alpha_t^2\sigma^2}{K} \qquad (12)
\end{aligned}
$$

where the second inequality comes from

$$
\frac{1}{K}\sum_{k\in[K]}\mathbb{E}[\|\bar{x}^{(t)} - x^{(t)}\|^2] \leq \frac{1}{K}\sum_{k\in[K]}\mathbb{E}[2\|x_k^{(t)} - \bar{x}^{(t)}\|^2 + 2\|\bar{x}^{(t)} - x^{(t)}\|^2]
$$

and

$$
\mathbb{E}[\|\bar{x}^{(t)} - x^{(t)}\|^2] = \frac{1}{K^2}\mathbb{E}[\|\sum_{k\in[K]}(x_k^{(t)} - x^{(t)})\|^2] \leq \frac{1}{K}\cdot\sum_{k\in[K]}\mathbb{E}[\|x_k^{(t)} - x^{(t)}\|^2].
$$

For simplicity, write $d_t = \mathbb{E}\left[\|\bar{x}^{(t)} - x^*\|^2\right]$, $e_t = \mathbb{E}\left[F(\bar{x}^{(t)}) - F(x^*)\right]$ and $\Xi_t = \frac{1}{K}\sum_{k\in[K]}\mathbb{E}[\|\bar{x}^{(t)} - x^{(t)}\|^2]$. Then we can rewrite equation 12 as

$$
d_{t+1} \leq (1 - \mu\alpha_t)\,d_t + \left(\frac{24L\alpha_t^2}{K^2} - \frac{\alpha_t}{2}\right)e_t + \left(8\alpha_t L + \frac{12\eta_t^2\alpha_t^2(1-p)}{pK}\right)\Xi_t + \frac{12\alpha_t^2\sigma^2}{K}.
$$

Multiplying both sides by $\frac{1}{\alpha_t}$ and rearranging, we have

$$
\left(\frac{1}{2} - \frac{24L\alpha_t}{K^2}\right)e_t \leq \frac{1 - \mu\alpha_t}{\alpha_t}d_t - \frac{1}{\alpha_t}d_{t+1} + \left(8L + \frac{12\eta_t^2\alpha_t(1-p)}{pK}\right)\Xi_t + \frac{12\alpha_t\sigma^2}{K}.
$$

From $\alpha_t = \alpha \leq \frac{p}{48HL} \leq \frac{1}{4L}$ and $\eta_t = \eta = \frac{p}{2H\alpha}$, we can further simplify the above inequality as

$$
\frac{1}{4}e_t \leq \frac{1 - \mu\alpha}{\alpha}d_t - \frac{1}{\alpha}d_{t+1} + 9L\Xi_t + \frac{12\alpha\sigma^2}{K}.
$$

Next, by taking the weighted average from $t = 0$ to $T - 1$ with weight $w_t = (1 - \mu\alpha)^{-(t+1)}$ and normalizing factor $W_T = \sum_{t=0}^{t-1} w_t$, we have

$$
\begin{aligned}
\frac{1}{4W_T}\sum_{t=0}^{T-1}w_t e_t &\leq \frac{1}{\alpha}\cdot\frac{1}{W_T}\sum_{t=0}^{T-1}((1 - \mu\alpha)w_t d_t - w_t d_{t+1}) \\
&\quad + \frac{9L}{W_T}\sum_{t=0}^{T-1}w_t\Xi_t + \frac{12\alpha\sigma^2}{K}\cdot\frac{1}{W_T}\sum_{t=0}^{T-1}w_t \\
&\leq \frac{1}{\alpha}\cdot\frac{1}{W_T}\sum_{t=0}^{T-1}(w_{t-1}d_t - w_t d_{t+1}) \\
&\quad + \frac{9L}{W_T}\sum_{t=0}^{T-1}w_t\Xi_t + \frac{12\alpha\sigma^2}{K}\cdot\frac{1}{W_T}\sum_{t=0}^{T-1}w_t \qquad (\text{By } (1 - \mu\alpha)w_t = w_{t-1}) \\
&\leq \frac{1}{\alpha}\cdot\frac{1}{W_T}\sum_{t=0}^{T-1}(w_{t-1}d_t - w_t d_{t+1})
\end{aligned}
$$

$$+ \frac{9L}{W_T} \cdot \left( 9\alpha^2 H^2 \sum_{t=0}^{T-1} w_t \sigma^2 + \frac{p^2}{2L} \sum_{t=0}^{T-1} w_t e_t \right) + \frac{12\alpha\sigma^2}{K} \qquad \text{(By Lemma 4)}$$

$$\leq \frac{1}{\alpha} \cdot \frac{1}{W_T} \left( d_0 - w_T d_{t+1} \right)$$

$$+ \frac{9L}{W_T} \cdot \left( 9\alpha^2 H^2 \sum_{t=0}^{T-1} w_t \sigma^2 + \frac{p^2}{48L} \sum_{t=0}^{T-1} w_t e_t \right) + \frac{12\alpha\sigma^2}{K}$$

(Taking telescoping sum on the first term)

Rearranging and using $p \leq \frac{1}{2}$, we get that

$$\frac{1}{5} \cdot \frac{1}{W_T} \sum_{t=0}^{T-1} w_t e_t \leq \frac{1}{\alpha} \cdot \frac{1}{W_T} \left( d_0 - w_T d_{t+1} \right) + \left( 81 L H^2 \alpha^2 + \frac{12\alpha}{K} \right) \sigma^2$$

$$\leq \frac{d_0}{\alpha} (1 - \mu\alpha)^T + \left( 81 L H^2 \alpha^2 + \frac{12\alpha}{K} \right) \sigma^2$$

$$\leq \frac{d_0}{\alpha} \exp(-\mu\alpha T) + \left( 81 L H^2 \alpha^2 + \frac{12\alpha}{K} \right) \sigma^2$$

Let

$$g(\alpha) = \frac{d_0}{\alpha} \exp(-\mu T \alpha) + 81 L \alpha^2 H^2 \sigma^2 + \frac{12\alpha\sigma^2}{K}$$

We wish to minimize $g(\alpha)$ with respect to $\alpha$. To do so, differentiate $g(\alpha)$ with respect to $\alpha$ we get

$$g'(\alpha) = -d_0 \exp(-\mu T \alpha) \left( \frac{1}{\alpha^2} + \frac{\mu T}{\alpha} \right) + 162 L H^2 \sigma^2 \alpha + \frac{12\sigma^2}{K}, \tag{13}$$

and we want to find $\alpha^*$ such that $g'(\alpha^*) = 0$. Since $g''(\alpha) > 0$, such $\alpha^*$ is the minimizer of $g(\alpha)$.

However, it is complicated to get the accurate value of $\alpha^*$. But since we will ignore the poly-logarithmic terms in our final bound of $g(\alpha)$, it suffices to approximate $\alpha^*$ within poly-logarithmic factor. This would give a proper approximation of the the second and the third terms of $g(\alpha^*)$. For the first term, we can use Equation equation 13 to derive the value of $\exp(-\mu T \alpha)$ and simplify the term to a polynomial of $\alpha$.

Suppose $\alpha^* = \frac{1}{\mu T} \ln A$. Our goal is to show that $A = \Theta(\text{poly}(T, H, K, \sigma^{-1}))$. First, by plugging $\alpha^* = \frac{1}{\mu T} \ln A$ to equation 13 we get

$$g'(\alpha^*) = -\frac{d_0}{A} \cdot \frac{(\ln A + 1)\mu^2 T^2}{\ln^2 A} + \frac{162 L H^2 \sigma^2 \ln A}{\mu T} + \frac{12\sigma^2}{K} = 0. \tag{14}$$

Simplifying both sides, we have

$$\frac{d_0}{A} \cdot \frac{(\ln A + 1)\mu^2 T^2}{\ln^2 A} = \frac{162 L H^2 \sigma^2 \ln A}{\mu T} + \frac{12\sigma^2}{K},$$

and

$$\frac{\ln A + 1}{A \ln^2 A} = \frac{\sigma^2}{d_0 \mu^2 T^2} \cdot \left( \frac{162\kappa H^2 \ln A}{T} + \frac{12}{K} \right).$$

Assuming $T$ to be sufficiently large and $\frac{162\kappa H^2 \ln A}{T} \ll \frac{12}{K}$, we can further simplify the above equality as

$$\frac{\ln A + 1}{A \ln^2 A} = \frac{C\sigma^2}{d_0 \mu^2 T^2 K} \tag{15}$$

for some constant $C > 12$. Notice that for any $A > 4$, $\frac{\ln^2 A}{\ln A + 1} \leq A$ and $\frac{\ln^2 A}{\ln A + 1} \geq \frac{1}{\sqrt{A}}$, we have

$$\frac{1}{\sqrt{A}} \leq \frac{\ln A + 1}{A \ln^2 A} \leq \frac{1}{A^2},$$

for any $A \geq 4$. Plugging back to equation 15, we have that

$$\frac{1}{\sqrt{A}} \leq \frac{C\sigma^2}{d_0 \mu^2 T^2 K} \leq \frac{1}{A^2}.$$

Therefore $A$ is polynomial in $T, H, K$ and $\sigma^{-1}$.

Now we upper-bound the minimal value of $g(\alpha)$. Since Theorem 2 also requires that $\alpha < \frac{p}{48LH}$, we need to discuss the following two cases:

If $\alpha^* < \frac{p}{48LH}$, then $g(\alpha)$ takes the minimum when $\alpha = \alpha^*$. Plugging in $\alpha^* = \frac{1}{\mu T} \ln A$ we have

$$
\begin{aligned}
g(\alpha) &= g(\alpha^*) \\
&= \frac{d_0}{\alpha^*} \exp(-\mu T \alpha^*) + 81L(\alpha^*)^2 H^2 \sigma^2 + \frac{12\alpha^* \sigma^2}{K} \\
&= \left(162LH^2\sigma^2\alpha^* + \frac{12\sigma^2}{K}\right) \cdot \frac{1}{\mu T} + 81LH^2\sigma^2(\alpha^*)^2 + \frac{12\sigma^2\alpha^*}{K} \\
&= \frac{162LH^2\sigma^2 \ln A}{\mu^2 T^2} + \frac{12\sigma^2}{\mu K T} + \frac{81LH^2\sigma^2 \ln^2 A}{\mu^2 T^2} + \frac{12\sigma^2 \ln A}{\mu K T} \\
&= \tilde{O}\left(\frac{\kappa H^2 \sigma^2}{\mu T^2} + \frac{\sigma^2}{\mu K T}\right).
\end{aligned}
$$

Here the step comes from equation 13, the third step come from $\alpha^* = \frac{1}{\mu T} \ln A$, and the last step comes from $A$ is polynomial in $T, H, K$ and $\sigma^{-1}$.

If $\alpha^* \geq \frac{p}{48LH}$, then since $g(\alpha)$ is convex in $\alpha$, it is monotonically non-increasing when $\alpha \leq \frac{p}{48LH}$. Thus $g(\alpha)$ takes the minimal value when $\alpha = \frac{p}{48LH}$. Therefore

$$g(\alpha) = g(\frac{p}{48LH}) \leq \tilde{O}(\frac{d_0 LH}{p} \exp(-\frac{\mu p T}{48LH}) + \frac{\kappa H^2 \sigma^2}{\mu T^2} + \frac{\sigma^2}{\mu K T})$$

Here we use $\frac{p}{48LH} \leq \alpha^* \leq \tilde{O}(\frac{1}{\mu T})$ on the second and the third terms in the second step.

Combining the two cases, we have

$$\frac{1}{5} \cdot \frac{1}{W_T} \sum_{t=0}^{T-1} w_t e_t \leq \left(\frac{d_0 LH}{p} \exp(-\frac{\mu p T}{48LH}) + \frac{\kappa H^2 \sigma^2}{\mu T^2} + \frac{\sigma^2}{\mu K T}\right)$$

The proof then follows from

$$\mathbb{E}[F(\hat{x}_T)] - F(x^*) \leq \frac{1}{W_T} \sum_{t=0}^{T-1} w_t \mathbb{E}[F(x^{(t)})] - F(x^*) \leq \frac{1}{W_T} \sum_{t=0}^{T-1} w_t e_t.$$

$\square$

## A.5 Proof of Main Technical Lemmas

### A.5.1 Proof of Lemma 2

*Proof.* Noting that $\{g_k^{(t)} - (x_k^{(t)} + x^{(t)})b_k^{(t)} - \nabla F(x_k^{(t)})\}_{k=1}^K$ are independent zero-mean random vectors, we have that for any $t \geq 0$,

$$\mathbb{E}\left[\|g^{(t)} - \bar{g}^{(t)}\|^2\right] = \mathbb{E}\left[\left\|\frac{1}{K}\sum_{k=1}^K \left(g_k^{(t)} - \nabla F(x_k^{(t)}) + \eta_t x^{(t)} - \eta_t x_k^{(t)}\right)\right\|^2\right]$$

$$= \frac{1}{K^2} \sum_{k=1}^{K} \mathbb{E} \left[ \left\| g_k^{(t)} - \nabla F(x_k^{(t)}) + \eta_t x^{(t)} - \eta_t x_k^{(t)} \right\|^2 \right]. \tag{16}$$

where the last term comes from the independence of $\{g_t^k\}$. Recall that

$$g_k^{(t)} = \frac{1 - b_k^{(t)}}{1 - p} \nabla f(x_k^{(t)}, \xi_k^{(t)}) + \frac{\eta_t b_k^{(t)}}{pK} (x_k^{(t)} - x^{(t)}).$$

where $b_k^{(t)}$ is a Bernoulli random variable that equals to 1 with probability $p$ and 0 with probability $1 - p$. Also $\xi_k^{(t)}$ denotes the data sampled by the $k$-th server at step $t$. Let $\mathcal{F}_t = \{\xi_k^{(s)}\}_{s=0,\cdots,t-1,k\in[K]} \cup \{b_k^{(s)}\}_{s=0,\cdots,t-1,k\in[K]}$. Conditional on any fixed $\mathcal{F}_t$, we then rewrite each of the summands as

$$\mathbb{E} \left[ \left\| g_k^{(t)} - \eta_t x_k^{(t)} + \eta_t x^{(t)} - \nabla F(x_k^{(t)}) \right\|^2 \right]$$

$$= \mathbb{E}_{b_k^{(t)}, \xi_k^{(t)}} \left[ \left\| g_k^{(t)} - \eta_t x_k^{(t)} + \eta_t x^{(t)} - \nabla F(x_k^{(t)}) \right\|^2 \right]$$

$$= \mathbb{E}_{b_k^{(t)}, \xi_k^{(t)}} \left[ \left\| \frac{1 - b_k^{(t)}}{1 - p} \nabla f(x_k^{(t)}, \xi_k^{(t)}) + \frac{\eta_t b_k^{(t)}}{pK} (x_k^{(t)} - x^{(t)}) - \eta_t x_k^{(t)} + \eta_t x^{(t)} - \nabla F(x_k^{(t)}) \right\|^2 \right]$$

$$= (1 - p) \cdot \mathbb{E}_{\xi_k^{(t)}} \left[ \left\| \frac{1}{1 - p} \nabla f(x_k^{(t)}, \xi_k^{(t)}) - \nabla F(x_k^{(t)}) - \eta_t x_k^{(t)} + \eta_t x^{(t)} \right\|^2 \right]$$

$$+ p \cdot \left\| \left( \frac{1}{p} - 1 \right) \eta_t (x_k^{(t)} - x^{(t)}) - \nabla F(x_k^{(t)}) \right\|^2$$

$$= \frac{1}{1 - p} \cdot \mathbb{E}_{\xi_k^{(t)}} \left[ \|\nabla f(x_k^{(t)}, \xi_k^{(t)}) - \nabla F(x_k^{(t)})\|^2 \right] + (1 - p) \left\| \eta_t x_k^{(t)} - \eta_t x^{(t)} - \frac{p}{1 - p} \nabla F(x_k^{(t)}) \right\|^2$$

$$+ \frac{(1 - p)^2}{p} \left\| \eta_t x_k^{(t)} - \eta_t x^{(t)} - \frac{p}{1 - p} \nabla F(x_k^{(t)}) \right\|^2$$

$$= \frac{1}{1 - p} \cdot \mathbb{E}_{\xi_k^{(t)}} \left[ \|\nabla f(x_k^{(t)}, \xi_k^{(t)}) - \nabla F(x_k^{(t)})\|^2 \right] + \frac{1 - p}{p} \cdot \left\| \eta_t (x_k^{(t)} - x^{(t)}) - \frac{p}{1 - p} \nabla F(x_k^{(t)}) \right\|^2$$

Notice that for all $k \in [K]$,

$$\mathbb{E}_{\xi_k^{(t)}} \left[ \|\nabla f(x_k^{(t)}, \xi_k^{(t)}) - \nabla f(x_k^{(t)})\|^2 \right]$$

$$\leq \mathbb{E}_{\xi_k^{(t)}} [\|\nabla f(x_k^{(t)}, \xi_k^{(t)})\|^2]$$

$$\leq \mathbb{E}_{\xi_k^{(t)}} [\|\nabla f(x_k^{(t)}, \xi_k^{(t)}) - \nabla f(x^*, \xi_k^{(t)}) + \nabla f(x^*, \xi_k^{(t)})\|^2]$$

$$\leq \mathbb{E}_{\xi_k^{(t)}} [\|\nabla f(x_k^{(t)}, \xi_k^{(t)}) - \nabla f_{\xi_k^{(t)}}(x^{(t)}) + \nabla f_{\xi_k^{(t)}}(x^{(t)}) - \nabla f(x^*, \xi_k^{(t)}) + \nabla f(x^*, \xi_k^{(t)})\|^2]$$

$$\leq 3 \cdot \mathbb{E}_{\xi_k^{(t)}} [\|\nabla f(x_k^{(t)}, \xi_k^{(t)}) - \nabla f_{\xi_k^{(t)}}(x^{(t)})\|^2 + \|\nabla f_{\xi_k^{(t)}}(x^{(t)}) - \nabla f(x^*, \xi_k^{(t)})\|^2 + \|\nabla f(x^*, \xi_k^{(t)})\|^2]$$

$$\leq 3L^2 \mathbb{E}[\|x_k^{(t)} - x^{(t)}\|^2] + 6L \mathbb{E}_{\xi_k^{(t)}} [f(x^{(t)}, \xi_k^{(t)}) - f(x^*, \xi_k^{(t)})] + 3\sigma^2$$

$$= 3L^2 \mathbb{E}[\|x_k^{(t)} - x^{(t)}\|^2] + 6L \mathbb{E}[F(x^{(t)}) - F(x^*)] + 3\sigma^2 \tag{17}$$

and from Jensen's inequality,

$$\|\nabla f(x_k^{(t)})\|^2 \leq \mathbb{E}_{\xi_k^{(t)}} \left[ \|\nabla f(x_k^{(t)}, \xi_k^{(t)})\|^2 \right] \leq 3L^2 \mathbb{E}[\|x_k^{(t)} - x^{(t)}\|^2] + 6L(F(x^{(t)}) - F(x^*)) + 3\sigma^2,$$

we then have

$$\mathbb{E}_{b_k^{(t)}, \xi_k^{(t)}} \left[ \frac{1}{1 - p} + \frac{1 - p}{p} \left\| \eta_t (x_k^{(t)} - x^{(t)}) - \frac{p}{1 - p} \nabla F(x_k^{(t)}) \right\|^2 \right]$$

$$\leq \frac{1}{1-p} \mathbb{E}_{\xi_k^{(t)}} \left[ \|\nabla f(x_k^{(t)}, \xi_k^{(t)}) - \nabla F(x_k^{(t)})\|^2 \right] + \frac{2\eta_t^2(1-p)}{p} \|x_k^{(t)} - x^{(t)}\|^2 + \frac{2p}{1-p} \|\nabla F(x_k^{(t)})\|^2$$

$$\leq \frac{(1+2p)}{1-p} \cdot (3L^2\|x_k^{(t)} - x^{(t)}\|^2 + 6L\left(F(x^{(t)}) - F(x^*)\right) + 3\sigma^2) + \frac{2\eta_t^2(1-p)}{p} \|x_k^{(t)} - x^{(t)}\|^2$$

$$\leq \frac{3\eta_t^2(1-p)}{p} \|x_k^{(t)} - x^{(t)}\|^2 + 24L\left(F(x^{(t)}) - F(x^*)\right) + 12\sigma^2.$$

Where the last step comes from $p \leq \frac{1}{2}$ and $12L^2 \leq \frac{\eta_t^2}{p}$. Substituting the last inequality into equation 16 and taking expectation over $\mathcal{F}_t$ yields the desired inequality. $\qquad\square$

### A.5.2 Proof of Lemma 3

*Proof.* Note that $x^{(t)} = \frac{1}{K}\sum_{k\in[K]} x_k^{(t^-)} = \bar{x}^{(t^-)} = x_k^{(t^-)}$. Here the last equality comes from that the client model synchronizes with the average model at step $t^-$. Therefore, we can write $\frac{1}{K}\sum_{k\in[K]} \mathbb{E}[\|x_k^{(t)} - x^{(t)}\|^2]$ as

$$\frac{1}{K}\sum_{k\in[K]} \mathbb{E}[\|x_k^{(t)} - x^{(t)}\|^2] = \frac{1}{K}\sum_{k\in[K]} \mathbb{E}[\|x_k^{(t)} - x_k^{(t^-)}\|^2] \tag{18}$$

Using $\|\sum_{i=1}^{H} x_i\|^2 \leq H \cdot \sum_{i=1}^{H} \|x_i\|^2$ we have

$$\frac{1}{K} \cdot \sum_{k\in[K]} \mathbb{E}\left[\|x_k^{(t)} - x_k^{(t^-)}\|^2\right]$$

$$\leq \frac{H}{K} \cdot \sum_{k\in[K]} \sum_{s=t^-}^{t-1} \mathbb{E}\left[\|x_k^{(t+1)} - x_t^k\|^2\right]$$

$$= \frac{H}{K} \cdot \sum_{k\in[K]} \sum_{s=t^-}^{t-1} \left(\frac{\alpha_s^2}{(1-p)} \cdot \mathbb{E}_{\xi_k^{(s)}}[\|\nabla f(x_k^{(s)}, \xi_k^{(s)})\|^2] + \frac{\alpha_s^2\eta_s^2}{p} \cdot \mathbb{E}[\|x_k^{(s)} - x^{(t^-)}\|^2]\right)$$

$$\leq \frac{H}{K} \cdot \sum_{k\in[K]} \sum_{s=t^-}^{t-1} \left(2\alpha_s^2 \cdot \left(6L\mathbb{E}[F(x^{(s)}) - F(x^*)] + 3\sigma^2\right) + \left(\frac{\alpha_s^2\eta_s^2}{p} + 6\alpha_s^2L^2\right) \cdot \mathbb{E}[\|x_k^{(s)} - x^{(t^-)}\|^2]\right)$$

$$\leq \frac{H}{K} \cdot \sum_{k\in[K]} \sum_{s=t^-}^{t-1} \left(6\alpha_s^2 \cdot \left(2L\mathbb{E}[F(x^{(s)}) - F(x^*)] + \sigma^2\right) + \frac{p^2}{2H^2} \cdot \mathbb{E}[\|x_k^{(s)} - x^{(t^-)}\|^2]\right) d$$

$$\leq 12LH \cdot \sum_{s=t^-}^{t-1} \alpha_s^2 \frac{1}{K} \sum_{k\in[K]} \mathbb{E}[F(x^{(s)}) - F(x^*)] + 6H^2\alpha_{t^-}^2\sigma^2 + \frac{p^2}{2H} \cdot \sum_{s=t^-}^{t-1} \frac{1}{K} \sum_{k\in[K]} \mathbb{E}[\|x_k^{(s)} - x^{(t^-)}\|^2]. \tag{19}$$

where the third step comes from equation 17 and $p \leq \frac{1}{2}$, and the fourth step comes from $\alpha_s^2\eta_s^2 = \frac{p^2}{4H^2}$ and $\alpha_s^2 \leq \frac{p^2}{24L^2H^2}$. The lemma then follows. $\qquad\square$

### A.5.3 Proof of Lemma 4

*Proof.* Substituting $\Xi_{t^-}, \cdots, \Xi_{t-1}$ on the right-hand-side of equation 11 by equation 11, we get

$$\Xi_t \leq \frac{p}{2H} \cdot \sum_{s=t^-}^{t-1} \Xi_s + 6H\alpha^2 \sum_{s=t^-}^{t-1} \sigma^2 + 12LH \cdot \sum_{s=t^-}^{t-1} \alpha^2 e_s$$

$$\leq \frac{p}{2H} \cdot \sum_{s=t^-}^{t-1} \left(\frac{p}{2H} \cdot \sum_{r=t^-}^{s-1} \Xi_r + 6H\alpha^2 \sum_{r=t^-}^{s-1} \sigma^2 + 12LH \cdot \sum_{r=t^-}^{s-1} \alpha^2 e_r\right)$$

$$+ 6H\alpha^2 \sum_{s=t^-}^{t-1} \sigma^2 + 12LH \cdot \sum_{s=t^-}^{t-1} \alpha^2 e_s$$

$$\leq \frac{p^2}{4H} \cdot \sum_{s=t^-}^{t-2} \Xi_s + \sum_{s=t^-}^{t-2} (3Hp\alpha^2 + 6H\alpha^2)\sigma^2 + 6H\alpha^2\sigma^2 + \sum_{s=t^-}^{t-2} (6LHp\alpha^2 + 12LH\alpha^2)e_s + 12LH\alpha^2 e_{t-1}.$$

Repeat the above step for $t - 1 - t^-$ times and using $\Xi_{t^-} = 0$ gets

$$\Xi_t \leq \sum_{s=t^-}^{t-1} 6H\alpha^2(1+p)(1 - \frac{p^{s-t^-}}{2^{s-t^-}})\sigma^2 + \sum_{s=t^-}^{t-1} 12LH\alpha^2(1+p)(1 - \frac{p^{s-t^-}}{2^{s-t^-}})e_s$$

$$\leq 9H^2\alpha^2\sigma^2 + \sum_{s=t^-}^{t-1} 18LH\alpha^2 e_s \tag{20}$$

Taking average of $\Xi_t$ from $t = 0$ to $T$ with weight $w_t = (1 - c\alpha)^t$, we get

$$\sum_{t=0}^{T} w_t \Xi_t \leq 9H^2\alpha^2 \sum_{t=0}^{T} w_t \sigma^2 + 18LH\alpha^2 \sum_{t=0}^{T} w_t \sum_{s=t^-}^{t-1} e_s$$

$$\leq 9H^2\alpha^2 \sum_{t=0}^{T} w_t \sigma^2 + 18LH\alpha^2 \sum_{t=0}^{T} e_t \sum_{s=t^-}^{t-1} w_s$$

$$\leq 9H^2\alpha^2 \sum_{t=0}^{T} w_t \sigma^2 + 18LH\alpha^2 \sum_{t=0}^{T} w_t e_t \sum_{s=t^-}^{t-1} (1 + \frac{p}{H})^{s-t^-}$$

$$\leq 9H^2\alpha^2 \sum_{t=0}^{T} w_t \sigma^2 + \frac{p}{16L} \sum_{t=0}^{T} w_t e_t$$

where the third step comes from $w_{t+1} \leq (1 + \frac{p}{H})w_t$, and the last step comes from $\alpha_t \leq \frac{p}{48HL}$. The lemma then follows. $\square$

# B  Experimental Details

Our code is available at https://github.com/Hiroki11x/Pseudo-Asynchronous-LocalSGD.

We conducted our experiments on three datasets: CIFAR-10, ImageNet-1K, and TinyStories. The CIFAR-10 and ImageNet-1K datasets were used for image classification tasks, while TinyStories was employed for language modeling experiments. We implemented the distributed algorithm (DDP, Local SGD, DiLoCo and PALSGD) on three different distributed systems, referred to as Cluster A, Cluster B, and Cluster C which have varying GPU configurations and interconnects, as detailed below.

## B.1  Settings

**Hardware Configurations:**

We utilized three types of clusters for our experiments:

**Cluster A**

- GPU: `NVIDIA Tesla T4 (16GB)` x 4
- GPU Bandwidth: 320.0 GB/s
- GPU Interconnect: PCIe with NUMA Node Interconnect (No NVLink)

**Cluster B**

- GPU: `NVIDIA Tesla V100 DGXS (32GB)` x 8
- GPU Bandwidth: 897.0 GB/s
- GPU Interconnect: NVLink, 150 GB/s per GPU

**Cluster C**

- GPU: `NVIDIA L40s (48GB)` x 4
- GPU Bandwidth: 864.0 GB/s
- GPU Interconnect: PCIe with NUMA Node Interconnect (No NVLink)

**Software and Library Configurations:**

All GPU clusters used the following software environment:

- Python: 3.11.6
- PyTorch: 2.3.1+cu121
- CUDA: 12.1
- CUDNN: 8902

**Workloads:**

- **Small CNN on CIFAR-10 (Preliminary Simulation Experiments)**: The CIFAR-10 dataset [3] is widely used in machine learning research, particularly for image recognition tasks. It contains 60,000 color images, each measuring 32×32 pixels, evenly distributed across ten distinct classes including airplanes, automobiles, birds, cats, deer, dogs, frogs, horses, ships, and trucks. The dataset consists of 50,000 training images and 10,000 test images, with each class represented by 6,000 images. We employed a small CNN model in Table B.1 and the CIFAR10 dataset. We investigate accuracy degradation across varying numbers of workers, including extremely large worker configurations. Due to limited computing resources typical of academic institutions, experiments were performed

---

[3] https://www.cs.toronto.edu/~kriz/cifar.html

on a simulated multi-worker environment using a single GPU and a relatively small-scale model. When controlling seeds and other conditions, such simulated experiments accurately represent actual distributed learning in terms of accuracy. However, execution time results are not comparable, as parallelization within a GPU differs fundamentally from distributed settings. Therefore, only accuracy results are reported.

- **VGG-16 on CIFAR-10**: We conducted experiments on CIFAR-10 using the VGG-16 architecture (Simonyan & Zisserman, 2014) to evaluate algorithm performance. Unlike prior experiments with smaller CNNs, this setup aims to leverage a deeper and more expressive model to assess the impact of algorithmic changes under realistic training conditions. The VGG-16 model we used follows the standard configuration defined in the PyTorch `torchvision.models` module, consisting of 13 convolutional layers followed by 3 fully connected layers. The convolutional layers use $3 \times 3$ filters with padding to preserve spatial resolution and are interleaved with ReLU activations and max-pooling layers for downsampling. After the convolutional blocks, the resulting feature maps are passed through an adaptive average pooling layer and flattened. The classifier consists of two fully connected layers with 4,096 units each (activated by ReLU and followed by dropout), and a final linear layer that maps to 10 output classes corresponding to the CIFAR-10 categories.

- **ResNet50 on ImageNet-1K**: The ImageNet-1K dataset [4] is a widely used benchmark for image classification tasks, providing a comprehensive test for evaluating model performance on large-scale visual recognition. We employed the ResNet50 model [5] with 25.6 million parameters to test the effectiveness of the PALSGD algorithm in distributed training settings [6]. The ResNet50 model used in this experiment features a total of 50 layers, which include a series of convolutional, batch normalization, and ReLU activation layers arranged in residual blocks. The input size is set to 224x224 pixels, a standard for ImageNet classification. Each residual block has a bottleneck structure that reduces the number of parameters while maintaining the accuracy of the model. The final fully connected layer produces 1,000 output logits corresponding to the 1,000 ImageNet classes.

- **GPT-Neo on TinyStories**: The TinyStories dataset (Eldan & Li, 2023) is designed for small-scale text generation tasks, providing a benchmark for language modeling performance on short narrative texts. We evaluated the PALSGD algorithm under distributed training conditions using the GPT-Neo model[7]. Experiments were conducted using two different model configurations: one with 8 million parameters and another with 125 million parameters.

    For the 8M parameter configuration, the GPT-Neo model features a maximum positional embedding size of 300, which allows the model to handle input sequences of up to 300 tokens. The hidden size is set to 128, defining the dimensionality of the model's internal representations. It employs 8 attention heads to capture diverse relationships among tokens through its self-attention mechanism, and it is built with 8 hidden layers to learn complex hierarchical patterns in the data.

    In contrast, the 125M parameter configuration is designed to handle longer sequences and capture more complex patterns. In this setup, the model uses a maximum positional embedding size of 1024, a hidden size of 768, 12 attention heads, and 12 hidden layers. This higher-capacity configuration enhances the model's ability to process and understand more intricate language patterns compared to the 8M version.

**Training Configuration:**

All experimental results, unless otherwise noted, refer to the hyper parameter configuration with the best results for that metric. The results of the ablation study are shown in Section E. The target loss shown in the training curve plots is based on the loss achieved by DDP within the predefined epoch budget.

- **Small CNN on CIFAR-10 (Preliminary Simulation Experiments)**: The number of workers ranged from 4 to 64, while synchronization intervals tested were 32. PALSGD was set with $p$ at

---

[4]https://www.image-net.org/download.php

[5]https://pytorch.org/vision/stable/models/generated/torchvision.models.resnet50.html

[6]We have followed official pytorch implementation and have not done any special data augmentation: https://github.com/pytorch/examples/blob/main/imagenet/main.py

[7]https://huggingface.co/docs/transformers/en/model_doc/gpt_neo

Table B.1: Architecture of the small CNN model used for simulation experiments

| Layer | Type | Configuration |
|---|---|---|
| 1 | Convolution | channels: $3 \to 32$, kernel size: 5, padding: 2 |
| 2 | Activation | ReLU |
| 3 | Max Pooling | kernel size: 2, stride: 2 |
| 4 | Convolution | channels: $32 \to 64$, kernel size: 5, padding: 2 |
| 5 | Activation | ReLU |
| 6 | Max Pooling | kernel size: 2, stride: 2 |
| 7 | Fully Connected | input: $64 \times 8 \times 8$, output: 1024 |
| 8 | Activation | ReLU |
| 9 | Fully Connected | input: 1024, output: 10 |

0.25. The outer optimizer used was Nesterov momentum, and the inner optimizer was momentum SGD. Inner learning rates of 0.01, 0.001, and 0.0001 were explored. Additionally, $\eta$ was fixed at 1, and outer learning rate included 1, 0.1, and 0.01. No weight decay was applied, and training was conducted for 100 epochs.

- **CIFAR10 Training on VGG-16**:
  We trained the model for 200 epochs using 4 GPUs on a single node on **Cluster C**. The local batch size was set to 128 per GPU, resulting in a global batch size of 512. No gradient accumulation was used. The model architecture was selected from the torchvision model, with `VGG-16` as the default. We used Momentum SGD as the inner optimizer with learning rates selected from $\{0.025, 0.05, 0.075\}$, and Nesterov Momentum SGD as the outer optimizer. The outer learning rate was chosen from $\{0.2, 0.4, 0.8, 1.2, 1.6\}$, with a warm-up period of 10 epochs followed by cosine annealing (`CosineAnnealingLR`) for scheduling. Weight decay was fixed at $1 \times 10^{-4}$. The synchronization interval $H$ was set to 128, and Local SGD variants started after 2049 iterations (approximately at epoch 40). We used the PALSGD algorithm with decoupled updates enabled. For PALSGD-style adaptive synchronization, we explored probabilistic synchronization with $p \in \{0.02, 0.05, 0.1\}$ and local step size $\eta \in \{0.1, 0.25, 0.5\}$.

- **ImageNet-1K Training on ResNet-50**:
  We trained the model for 90 epochs with a local batch size of 64 per GPU, using 4 GPUs on **Cluster C**. The global batch size was set to 256. The inner optimizer's learning rate was fixed at 0.001 with the Momentum SGD. For outer optimizer, we used Nesterov Momentum SGD with outer learning rate fixed at 0.1 to 0.2 as same as GPT-Neo experiments. The synchronization interval $H$ was set to 64. The variants of the Local SGD algorithm started after 200K iterations (39 epoch). For PALSGD experiments, the probabilistic synchronization parameter p = 0.05 and $\eta_t$ is 0.5 to 16.

- **TinyStories Training on GPT-Neo-8M and 125M**:
  Both the 8M and 125M models were trained following the training protocol described below. We trained the model for 15 epochs with a local batch size of 512 per GPU, and use **Cluster A** in the experiments with 4 GPUs, and **Cluster B** in the experiments with 8 GPUs. The global batch size was set to 2048. The inner optimizer learning rate was fixed at 0.001, and we employed AdamW (Loshchilov, 2017) for the inner optimizer with gradient clipping enabled. Regarding outer optimization for DiLoCo and PALSGD, we use Nesterov Momentum SGD with the outer learning rate fixed at 0.1 to 0.2. The synchronization interval $H$ was set to 16 (125M) or 64 (8M), the probabilistic synchronization parameter $p = 0.1$, $\eta_t = 16$. The variants of the Local SGD algorithm started after 1024 iterations. For ablation study, the synchronization interval $H$ is set to 32 to 256, the probabilistic synchronization parameter $p = 0.025$ to 0.5 and $\eta_t$ is 0.25 to 64.

# C    Full Results of Experiments

### C.1    ImageNet-1K on ResNet50

Figure C.1 presents the training accuracy curves for ResNet-50 on ImageNet-1K with $K = 4$ workers and a synchronization interval of $H = 64$, conducted on **Cluster C**. Consistent with our validation accuracy results, PALSGD is 15.4% faster than DDP and 4.3% faster than DiLoCo, while all methods converge to the same final accuracy of 74.0%.

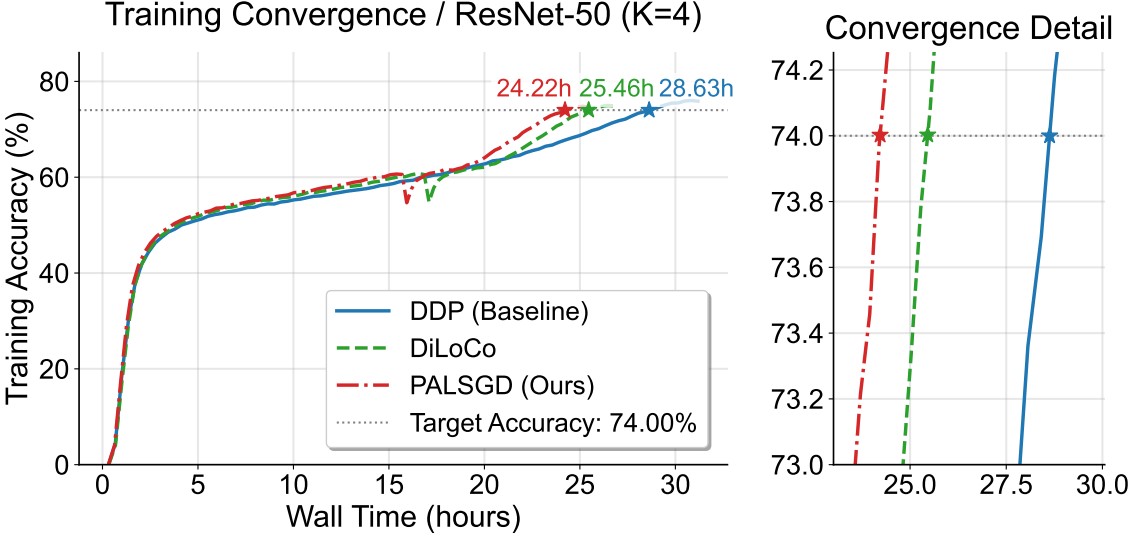

Figure C.1: Training Accuracy of ImageNet-1K on ResNet-50 Experiments ($K = 4$ / $H = 64$): PALSGD demonstrates faster training compared to DDP and DiLoCo.

## C.2    TinyStories on GPT-Neo-8M

### C.2.1    GPT-Neo-8M Experiment with 4 GPUs (K=4)

We conducted an additional experiment using GPT-Neo-8M with 4 workers on **cluster A**. Figure C.2 presents both the training and validation loss curves for this setup. Consistent with previous findings, PALSGD achieved the fastest convergence, reaching the target loss significantly earlier than both DiLoCo and DDP. Specifically, PALSGD reduced training time by 23.0% compared to DDP, while DiLoCo provided a more modest 7.9% improvement over DDP. DDP exhibited the slowest convergence, taking the longest time to reach the target loss. While DiLoCo was faster than DDP, it failed to reach the target loss within the given training time.

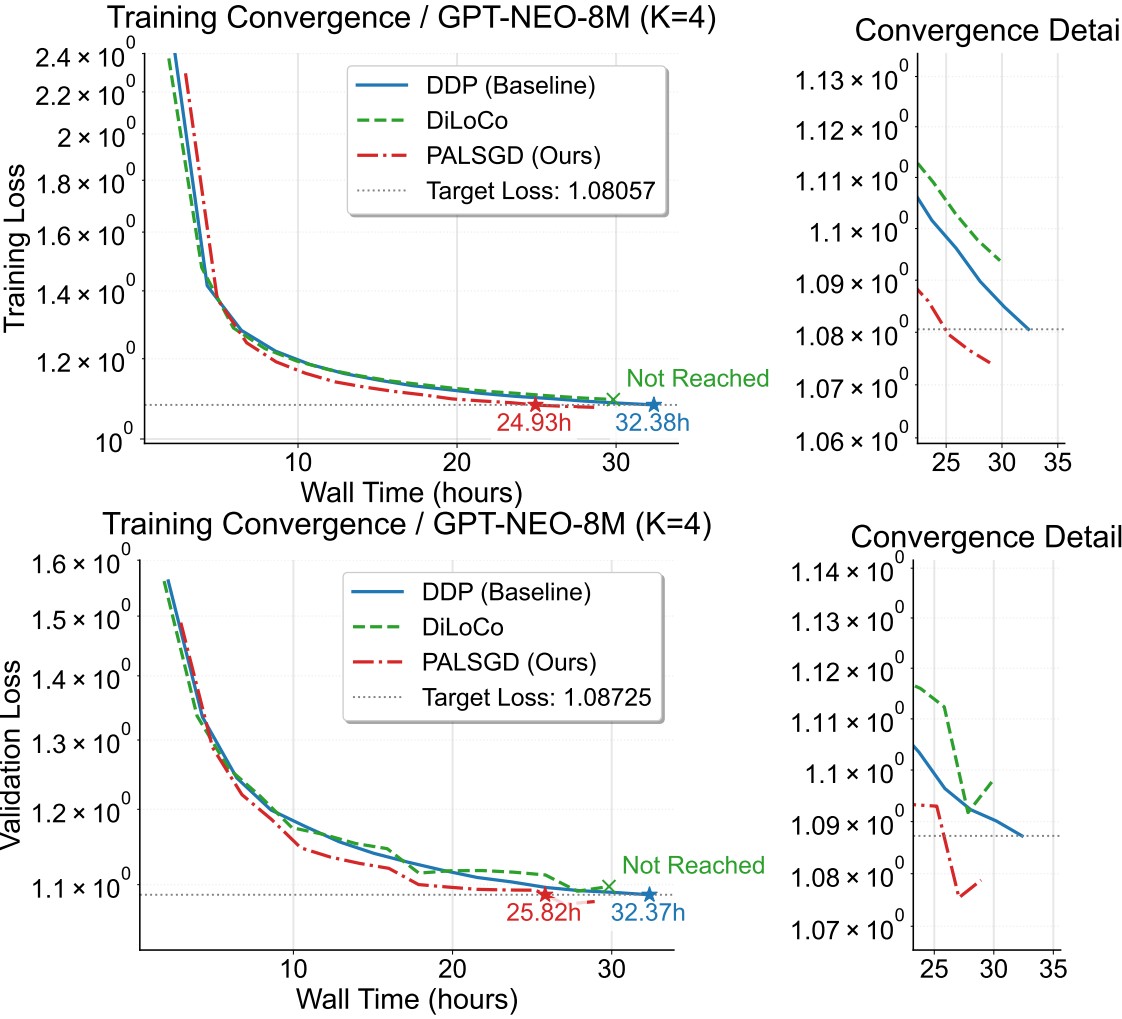

Figure C.2: GPT-Neo Experiments (K=4 / H=16): Training time comparison across distributed algorithm to achieve target training loss (top) and validation loss (bottom). PALSGD achieves fastest convergence and lowest loss, while DDP is slowest and DiLoCo did not achieve target loss.

### C.2.2 GPT-Neo-8M Experiment with 8 GPUs (K=8)

Figure C.3 presents the training loss curve for the GPT-Neo experiment with 8 workers on Cluster B. Consistent with our validation loss results, PALSGD demonstrates the fastest convergence, reaching the target training loss 20.8% faster than DDP. Among the three methods, DDP converges the slowest, while DiLoCo does not achieve the target loss within the given training time.

We remark that while it might seem that the communication bottleneck would be larger due to the doubled number of GPUs compared to the 4-GPU cluster, this is not necessarily the case. Since the GPUs in Cluster B do not share the same bandwidth and FLOPS as those in Cluster A, furthermore, Cluster B differs in that the GPUs are connected via NVLink, which provides faster communication. This makes it an ideal computing environment for distributed deep learning. As a result, we only achieved a 20% improvement in training speed. What we want to emphasize here is that even in an optimal communication environment like NVLink, which does not span across nodes, there is still room for a 20% increase in training speed.

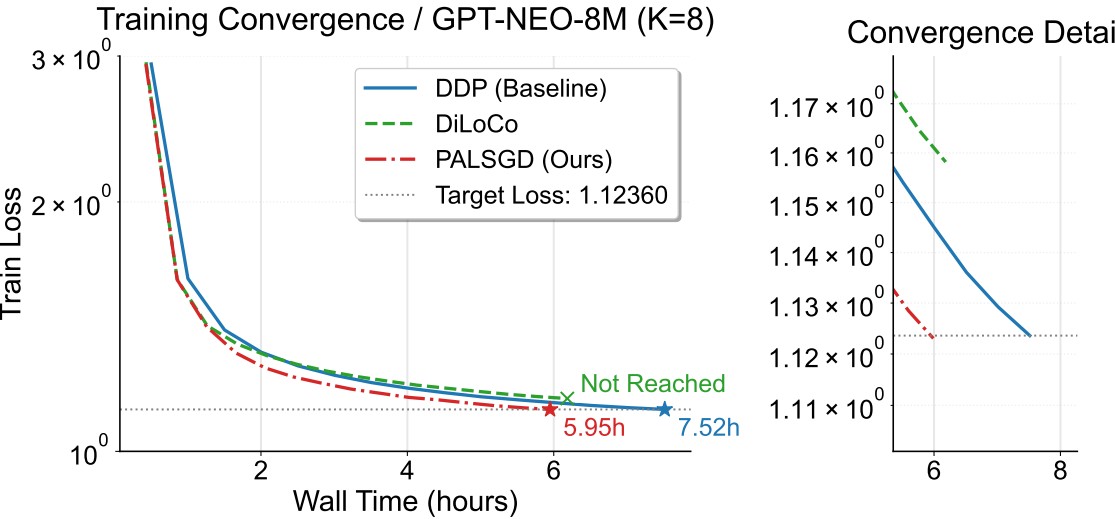

Figure C.3: GPT-Neo Experiments (K=8 / H=64) on DGX-1 (8 V100 GPUs Connected by NVLINK): Training time comparison across distributed algorithm to achieve target loss.

### C.3 TinyStories on GPT-Neo-125M

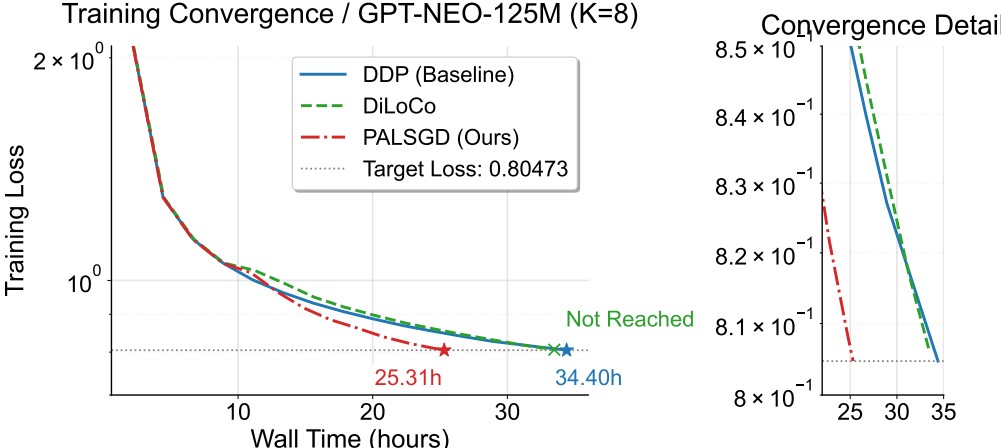

Figure C.4: GPT-Neo-125M Experiments (K=8 / H=16) Training time comparison across distributed algorithm to achieve target training loss.

Figure C.4 compares the training time required by PALSGD, DiLoCo, and DDP to reach a common target training. PALSGD and DDP eventually achieve the target loss, whereas DiLoCo fails to reach this target. Furthermore, PALSGD converges significantly faster than DDP. Specifically, PALSGD achieves the target training loss 26.4% faster than DDP. These results highlight PALSGD's superior efficiency in accelerating training while maintaining model quality.

### C.4  CIFAR-10 on VGG-16

We conducted experiments on the CIFAR-10 dataset using the VGG16 architecture Simonyan & Zisserman (2014) on cluster A with $k = 4$ workers. Figure C.5 presents the training loss and validation accuracy of the DDP, DiLoCo, and PALSGD algorithms for varying synchronization intervals ($H$). As expected, the results show an inverse relationship between $H$ and regularization strength for DiLoCo and PALSGD—i.e., training loss tends to be lower when using DDP or smaller values of $H$. The relatively poor performance of DDP is likely due to its use of large batch sizes. In contrast, DiLoCo and PALSGD operate with batch sizes reduced by a factor of four, which may result in a stronger implicit regularization effect. Compared to DiLoCo, PALSGD consistently achieves lower training loss and higher validation accuracy, suggesting better generalization performance. Moreover, PALSGD is less sensitive to the choice of $H$, showing only a 0.058% drop in accuracy when increasing $H$ from 32 to 1024. These results demonstrate the effectiveness of PALSGD in reducing communication overhead without compromising model performance.

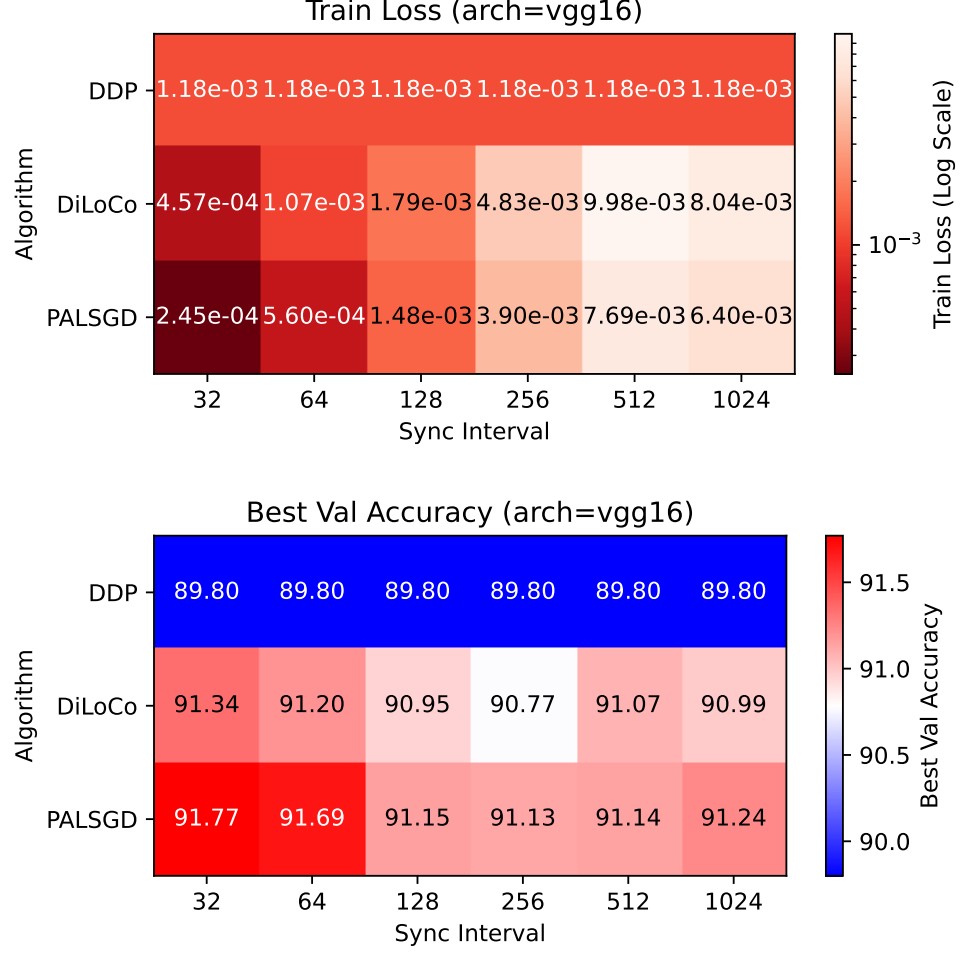

Figure C.5: CIFAR-10 with VGG16 Experiments: (Top) Comparison of training loss with respect to H (Sync Interval); (Bottom) Comparison of validation accuracy with respect to H. PALSGD achieves lowest loss and highest accuracy consistently.

# D Convergence Analysis by Global Steps

## D.1 Image Classification Tasks: ImageNet-1K on ResNet 50

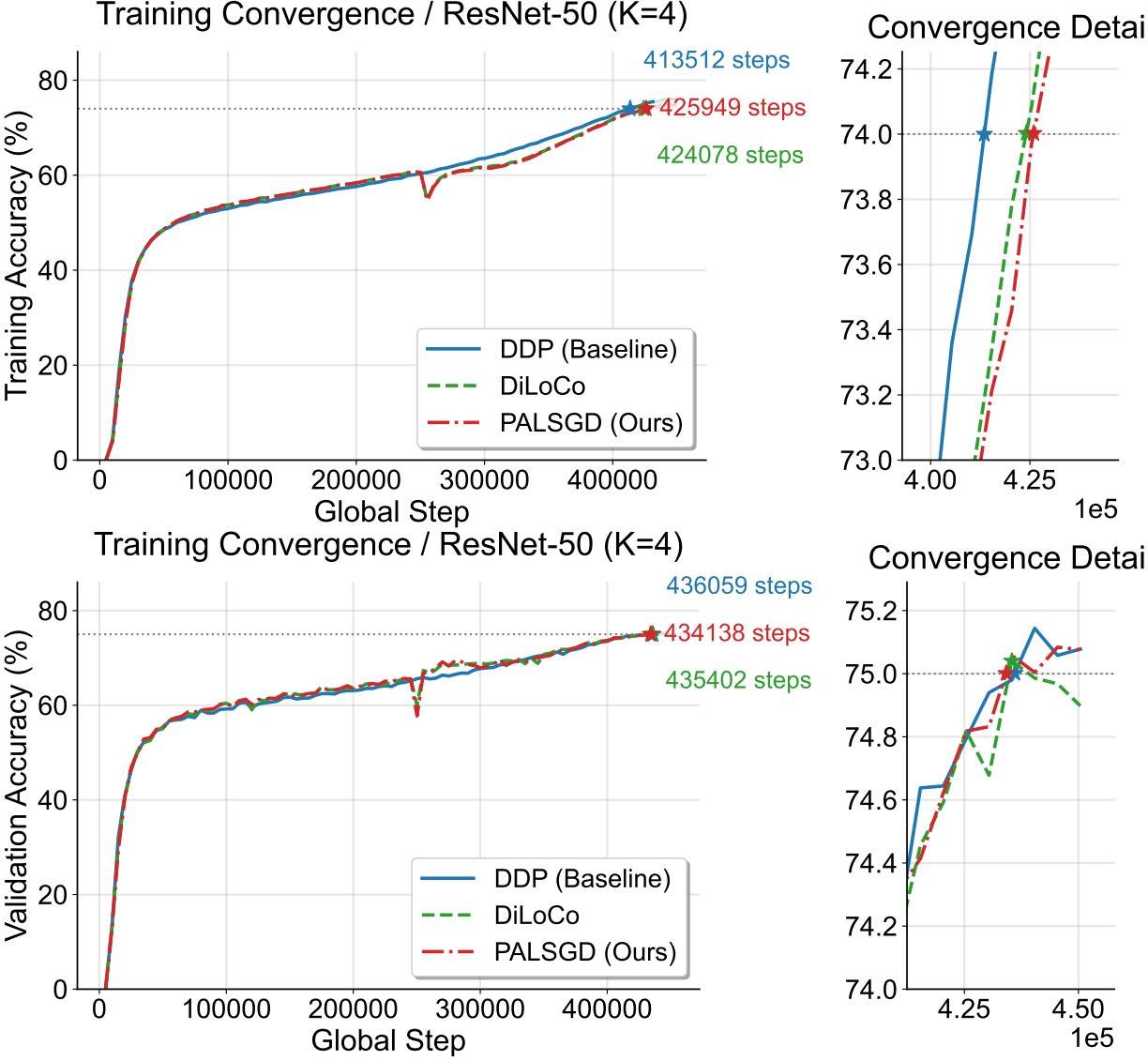

Figure D.1: Training (Top) and Validation (Bottom) Accuracy of ImageNet-1K on ResNet-50 Experiments ($K = 4$ / $H = 64$) with using Cluster C.

Our main results in Figure 2 present wall-clock time comparisons. To isolate the source of performance gains, this section provides a complementary analysis. We compare methods based on the number of global steps required to reach a target metric. This approach distinguishes algorithmic convergence speed from system-level speedups due to reduced communication. Image Classification on ImageNet-1K For the ResNet-50 experiment, all three methods—PALSGD, DiLoCo, and DDP—reached a given validation accuracy in a nearly identical number of global steps, as shown in Figure D.1 (bottom). DDP demonstrated a marginal advantage in the convergence of training accuracy. These results suggest the convergence properties of the algorithms are comparable for this task. The practical wall-clock advantage of PALSGD shown in Figure 2 therefore stems primarily from its reduction in communication overhead, not from superior step-wise convergence.

### D.2 Language Modeling Tasks: TinyStories on GPT-Neo

### D.2.1 GPT-Neo-8M Experiments

The GPT-Neo-8M experiments show that PALSGD and DDP again achieve similar convergence for both training and validation loss. In contrast, DiLoCo failed to converge, with its loss increasing during training. PALSGD's advantage over DDP on this model is its communication efficiency. Its advantage over DiLoCo is its stable convergence.

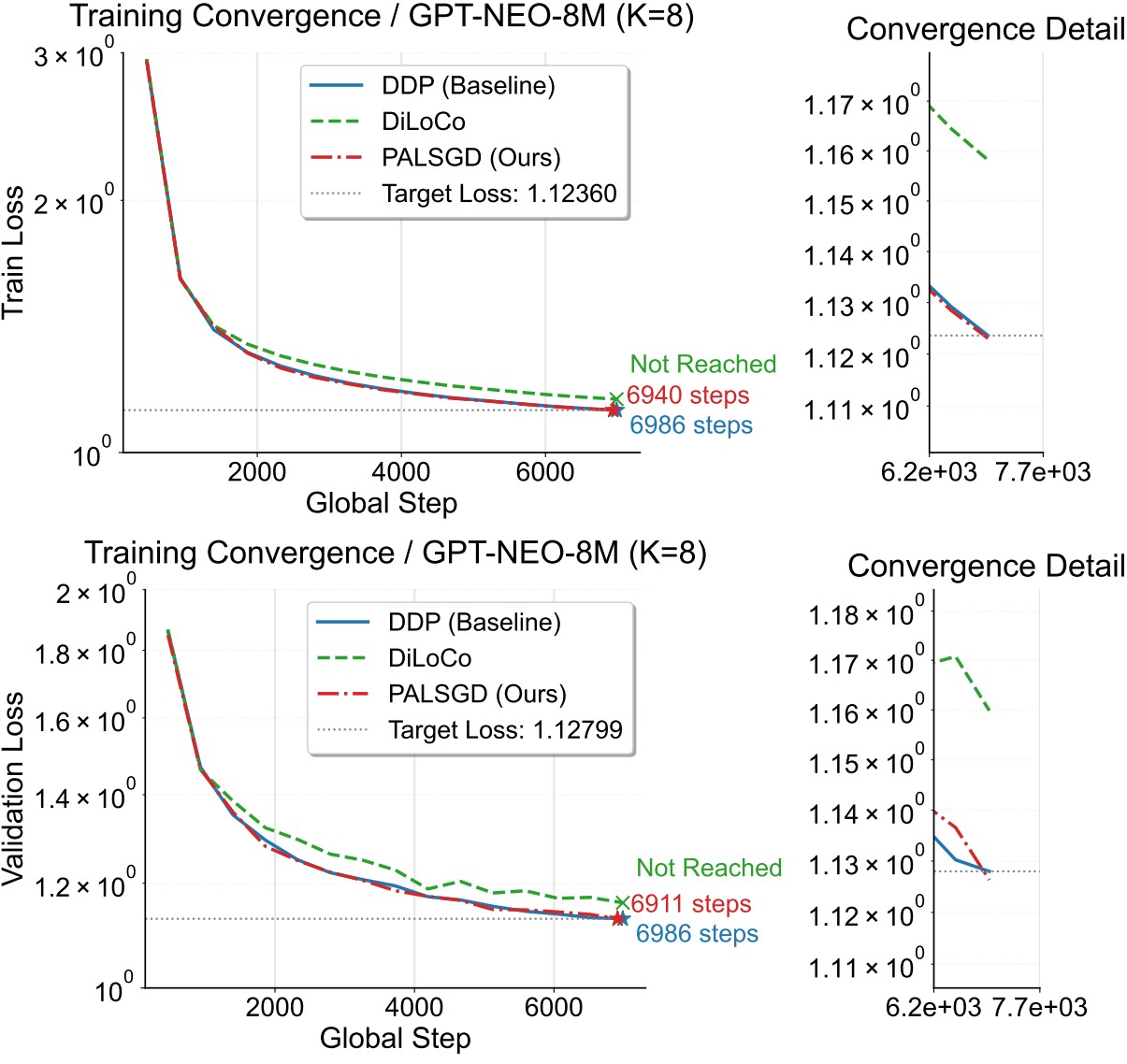

Figure D.2: GPT-Neo 8M Experiments ($K = 8$ / $H = 64$) on DGX-1 (8 V100 GPUs Connected by NVLINK): Training time comparison across distributed algorithm to achieve target loss.

### D.2.2 GPT-Neo-125M Experiments

Experiments with the larger GPT-Neo-125M model reveal a different dynamic. Here, PALSGD achieves the target loss in fewer steps than both DDP and DiLoCo. PALSGD required 15.8% fewer steps than DDP to reach the target validation loss. DiLoCo again failed to converge. This algorithmic improvement contributes significantly to the overall performance gain. The 24.4% wall-clock time reduction reported in Figure 3 is a product of both faster algorithmic convergence and reduced communication costs.

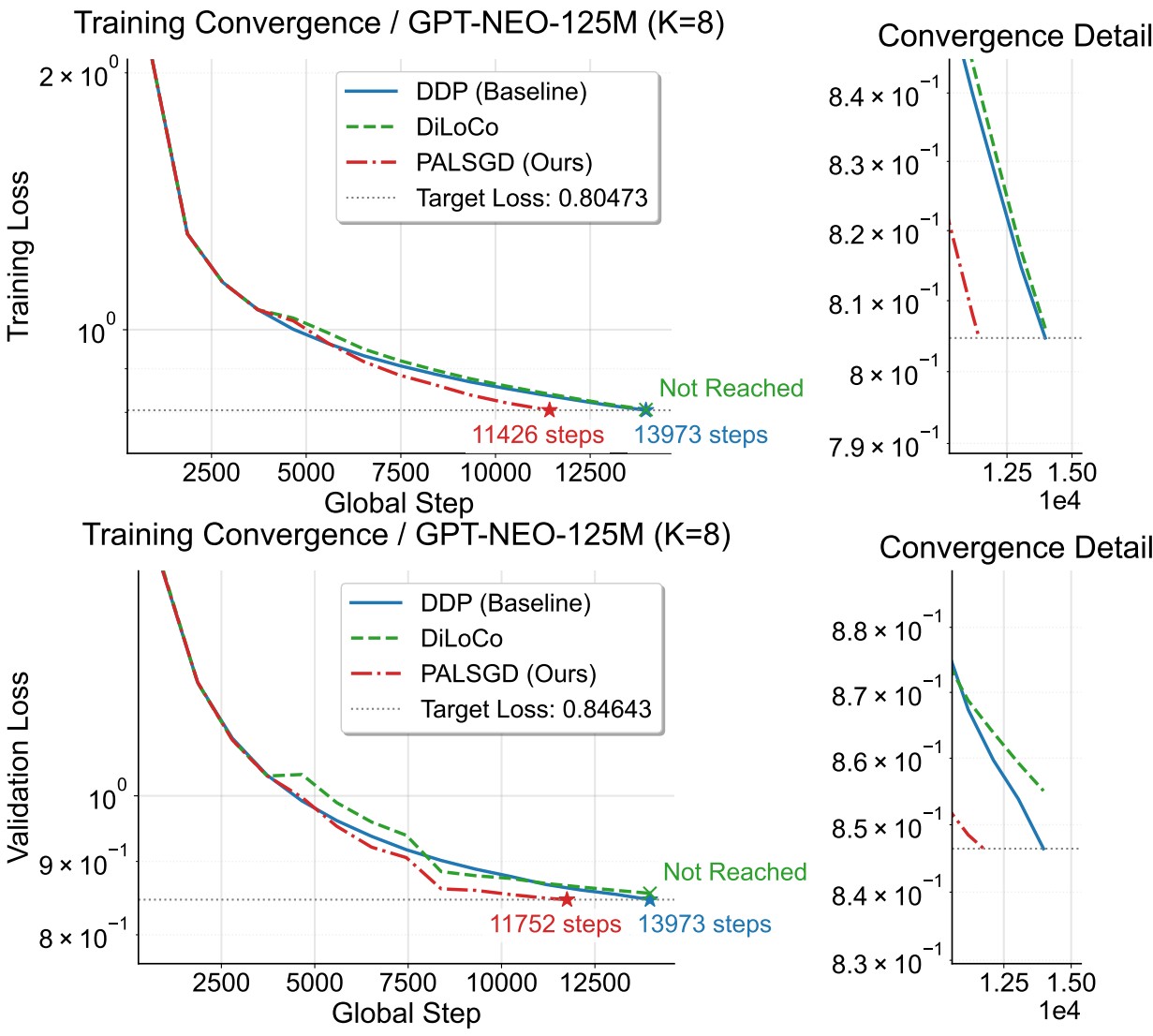

Figure D.3: GPT-Neo 125M Experiments ($K = 8$ / $H = 16$) on DGX-1 (8 V100 GPUs Connected by NVLINK): Training time comparison across distributed algorithm to achieve target loss.

# E Ablation Studies

Here, we present the results of an ablation study on hyperparameters using GPT-Neo-8M to train on the TinyStories dataset with K=8. We conducted ablation studies on the following hyperparameters: seeds, optimizer selection, $\eta$ , $p$, and $H$. Default hyperparameter setting is described in Appendix B.

The seeds represent the error bars in our experiments. While optimizer selection and $\eta$ influence the final loss and accuracy, they have minimal impact on training time, so we focused on evaluating these parameters using accuracy and loss metrics. On the other hand, $p$ and $H$ affect training time, so we report the time taken in addition to accuracy and loss. The result of the ablation study for $H$ is already included in Section 7.4.

Unless otherwise specified, the experiments follow a scheme in which only one hyperparameter is altered, with the rest set to their optimal values. Detailed descriptions of the hyperparameters are provided in the previous section.

## E.1 Sensitivity of Seeds

We evaluated different algorithms using three random seeds (2022, 2023, 2024). The results showed that the variance was minimal, confirming that it had no significant impact on the performance of the training process within the scope of our workload.

| Algorithm | Mean Validation Loss | Variance |
|-----------|---------------------|----------|
| DDP | 1.1247 | 1.0047e-05 |
| PALSGD | 1.1273 | 5.7668e-07 |
| DiLoCo | 1.1646 | 4.6850e-05 |

Table E.1: TinyStories on GPT-Neo-8M/ Mean and Variance of Final Validation Loss for PALSGD with Different Optimizer Combination

## E.2 Optimizer Selection for Loss

In line with the findings suggested by Douillard et al. (2023) for language modeling workloads, the combination of using AdamW as the inner optimizer and either Nesterov momentum or SGD with momentum as the outer optimizer not only achieved the best loss, but also revealed that using AdamW as the outer optimizer significantly worsened the loss.

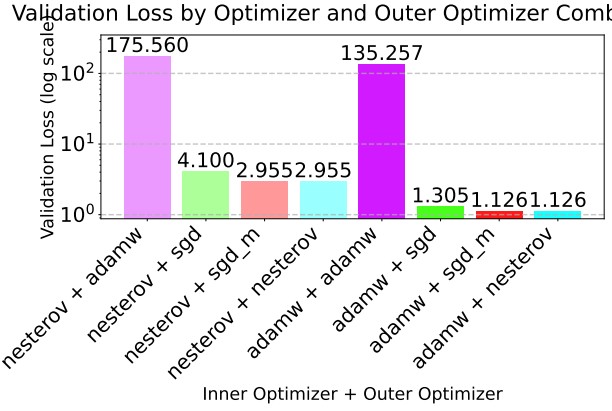

Figure E.1: Comparison of inner and outer optimizer with GPT-Neo Experiments on Cluster B (K=8, H=64). 'sgd_m' stands for SGD with Momentum

### E.3 Optimizer Selection for Convergence

Figure E.2 compares the time required for different outer optimizers to reach the target validation loss. The results show that the Nesterov optimizer achieves the shortest training time. In contrast, using SGD as the outer optimizer is significantly slower than using either Nesterov or SGD Momentum. Notably, when the outer optimizer is SGD, our algorithm reduces to a randomized variant of Local SGD.

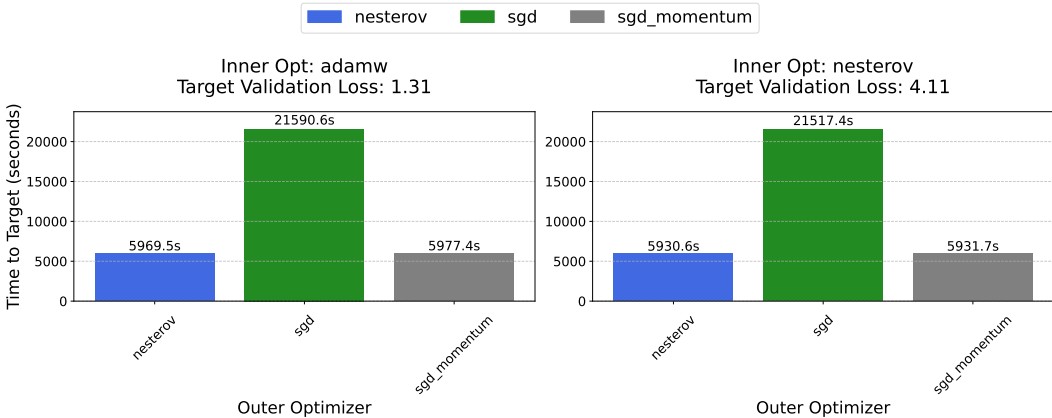

Figure E.2: Comparison of inner and outer optimizer with GPT-Neo Experiments on Cluster B (K=8, H=64).

### E.4 $\eta$ Selection

$\eta$ is a parameter that controls the update range in pseudo-synchronization, and larger values can be interpreted as stronger regularization. The experimental results showed that as eta increases, the validation loss also deteriorates significantly, likely due to excessive regularization. Additionally, it was found that when eta is below 4, no substantial changes occur.

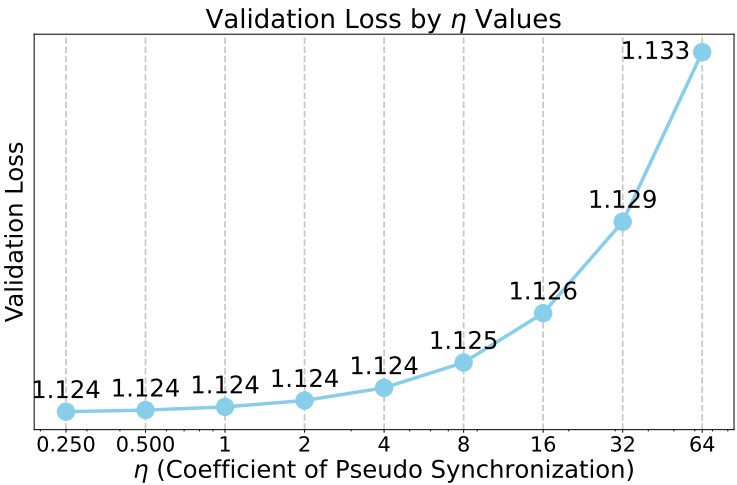

Figure E.3: Comparison of $\eta$ on GPT-Neo Experiments on Cluster B (K=8, H=64)

### E.5 $p$ Selection

The parameter $p$ controls the frequency of pseudo synchronization. When $p$ is too large, updates without gradients become more frequent, leading to insufficient progress in learning. However, since this skips forward and backward processes, the same number of steps is reached in a shorter amount of time. This intuitive trade-off aligns well with our experimental results.

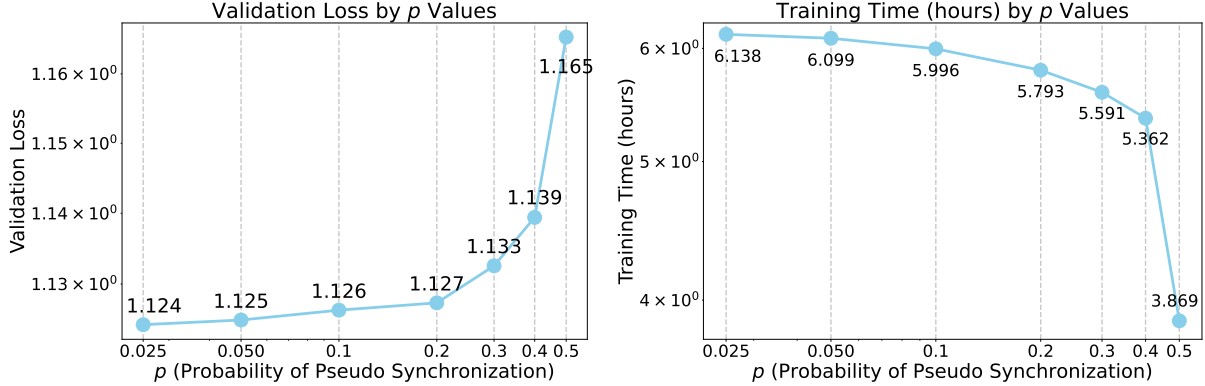

Figure E.4: Comparison of $p$ / GPT-Neo Experiments on Cluster B (K=8, H=64)

# F    Additional Study

## F.1    CIFAR10 Experiments on Scaling Performance

In this section, we conduct distributed training experiments using multiple nodes on Cluster C, rather than the simulated environment described in the previous section. PALSGD, DiLoCo, and DDP were evaluated using the CIFAR-10 dataset on the VGG-16 architecture. Experiments were executed on Cluster C with configurations of 1, 2, 4 and 8 nodes corresponding to 4, 8, 16 and 32 GPUs, respectively.

For optimization, local models utilized SGD with momentum. The outer optimizer was Nesterov momentum. Workers synchronized parameters every 128 steps. The local SGD phase began after 2048 global iterations, except for the K=16, 32 setting, where it started after 1024, 512 global iterations.

A grid search determined key hyperparameters. Inner learning rates were selected from {0.025, 0.05, 0.075}. Outer learning rates ranged across {0.2, 0.4, 0.8, 1.2, 1.6}. Algorithm-specific parameters included a gradient-drop probability $p$ selected from 0.02, 0.05, 0.1 and an eta coefficient chosen from {0.1, 0.25, 0.5}.

Several parameters remained constant. The batch size per GPU was set at 256, with no learning rate decay applied throughout training. To ensure high computational efficiency, we used the largest possible per-GPU batch size. The total number of training epochs was fixed for all experiments. This setup means that as the number of GPUs increases, the global batch size grows proportionally, while the total number of training steps decreases.

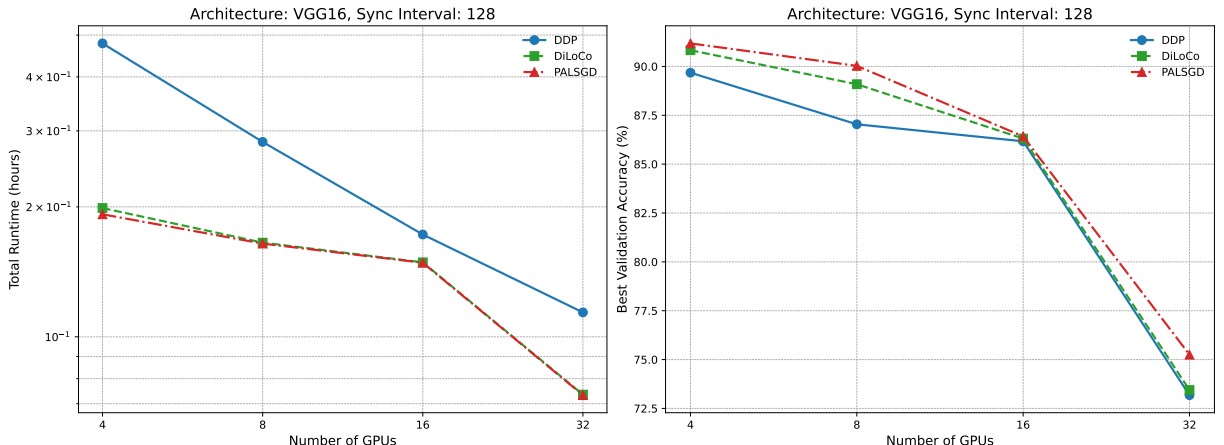

Figure F.1: Comparison of $K$ / CIFAR10 Experiments on Cluster C (K=4, 8, 16, 32, H=128)

Our baseline, DDP, showed a 4% drop in validation accuracy when scaled from 4 to 16 GPUs. The degradation was more severe at 32 GPUs, where accuracy fell by over 15%. This sharp decline on 32 GPUs is likely attributable to the large-batch problem on CIFAR-10. The global batch size becomes 8,192, which means an entire epoch over the 50,000-image dataset completes in just over six steps. Training time for all methods decreased linearly on a log scale, indicating good system scalability. PALSGD and DiLoCo have nearly identical runtimes. Their advantage in training time becomes more significant when scaling from 16 to 32 GPUs. This result suggests that both methods effectively mitigate the communication bottleneck that emerges at larger scales. However, they also suffer from accuracy degradation at 32 GPUs. Despite this, both methods consistently achieve higher validation accuracy than DDP. More importantly, PALSGD demonstrates superior accuracy to DiLoCo across all scales.

