# OpenReview forum: "Pseudo-Asynchronous Local SGD: Robust and Efficient Data-Parallel Training"
_TMLR — Accepted by TMLR_

### Review · Reviewer_oaZf · 2025-04-29

**Summary Of Contributions:**

1. The paper introduces PALSGD, an extension of Local SGD that incorporates probabilistic pseudo-synchronization.
2. The paper provided theoretical convergence bounds for a simplified version of PALSGD ie., µ-Strongly Convex case, the inner optimizer is standard SGD and the outer model is updated by taking the average across inner models.
3. Experiments were conducted on two different datasets: 1) ImageNet-1K with ResNet-50, and 2) TinyStories using GPT-Neo1 with 8 million and 125 million parameters. ImageNet was trained with 4 workers where as TinyStories was trained with 4 to 8 workers. PALSGD is compared against DDP and DiLoCo in terms of wall clock time to reach the same performance (Accuracy or loss). The experiments show that PALSGD reach the same performance faster as compared to DDP and DiLoCo.

**Audience:**

Yes

**Claims And Evidence:**

Yes

**Requested Changes:**

1. How does PALSGD compare to Scaffnew[1]? Since Scaffnew claims better performance than local-SGD, it might be insightful to compare PALSGD with Scaffnew.
2. The abstract highlights speedup with respect to DDP. However, the most competitive baseline in the current experiments is DiLoCo and therefore the speedup should be reported wrt DiLoCo (second best method). Is there a reason to pick DDP over DiLoCo for speedup computation?

Reference:
1. Mishchenko, Konstantin, et al. "Proxskip: Yes! local gradient steps provably lead to communication acceleration! finally!." International Conference on Machine Learning. PMLR, 2022.

**Strengths And Weaknesses:**

Strengths:
1. The paper address an important problem of communication bottleneck with large-scale training.
2. The paper processes an extension of local-SGD to improve the total training time required to reach a given performance.
3. The papers has exhaustive related work session.
4. The experimental results show that the proposed methods has  6% speedup over DiLoCo (ImageNet use-case).
5. Paper is well written and easy to follow.

Weaknesses/questions:
1. The paper motivates communication bottleneck for large-scale training but conducts experiments on 4-8 workers.
2. Theoretical analysis assumes strongly convex functions.
3. Section D.4 is not clear to me.  Can you please elaborate the setup used for both the figures?

---

> ### Author Response · Authors · 2025-06-16
> **Response to reviewer oaZf (1/n)**
>
> We thank the reviewer for their detailed feedback and insightful comments. We appreciate the acknowledgment of our work's contributions and find the suggestions very helpful for improving our paper. Below, we address each of the points raised.
>
> > The paper motivates communication bottleneck for large-scale training but conducts experiments on 4-8 workers.
>
>
> We appreciate the reviewer highlighting concerns about the scale of our experimental validation. To directly address this, we conducted additional experiments extending up to __32 GPUs (8 compute nodes)__ using CIFAR-10 with the VGG-16 architecture. See Appendix F in our revised version for detail.
>
> Due to practical constraints during the rebuttal period, we addressed the reviewer’s concerns through additional experiments using CIFAR-10. The original TinyStories experiments were conducted on a DGX-1 system, which limited us to a single node (8 GPUs). Scaling these experiments to multiple nodes (16 or 32 GPUs) would have required rerunning all configurations, entailing several hours per experiment and extensive computational resources for hyperparameter searches. Such computational resources were unavailable within the timeframe, barring perhaps exceptional scenarios for industry or larger research institutions.
>
> Similarly, the ImageNet experiments, performed on Cluster C (equipped with NVIDIA L40s x 4 GPUs), required more than 120 GPU hours per experiment. Reliable scaling experiments for ImageNet within the two-week rebuttal window were therefore unfeasible for similar reasons.
>
> Consequently, we selected CIFAR-10 for our additional experiments because it allowed us to complete individual experiments within approximately 0.5 GPU hours even at the 32-GPU (8-node) scale.
>
> In these extended experiments, the global batch size increased proportionally with GPU count, while total training epochs were constant across scales. At the 32-GPU scale, significant communication overhead was evident, confirming our setup effectively captures large-scale communication bottlenecks (Figure 18).
>
> Results clearly indicate that both PALSGD and DiLoCo outperform DDP, particularly at higher scales. Specifically, at 32 GPUs, PALSGD and DiLoCo significantly reduce runtime relative to DDP, validating their effectiveness in mitigating communication issues. Despite some accuracy degradation at this scale due to increased batch sizes, PALSGD consistently achieves higher validation accuracy compared to both DiLoCo and DDP across all evaluated scales.
>
> Thus, these expanded experiments robustly support our claims regarding PALSGD’s advantages in large-scale training scenarios, addressing the reviewer's concerns by demonstrating clear scalability improvements beyond the initially tested configurations.

---

> ### Author Response · Authors · 2025-06-16
> **Response to reviewer oaZf (2/2)**
>
> > Theoretical analysis assumes strongly convex functions.
>
> Thank you for this comment. You are correct that our theoretical analysis assumes strongly-convex functions, while our experiments are on non-convex deep learning tasks. This is a common simplification in the theoretical analysis of optimization algorithms, intended to provide initial, tractable insights into the algorithm's convergence behavior. Our paper's primary focus is on the practical utility and empirical performance of PALSGD. As we state in our "Limitations and Future Work" section, extending the theoretical framework to the non-convex setting is an important next step, but it is a substantial undertaking that we leave for future work.
>
> > Section D.4 is not clear to me. Can you please elaborate the setup used for both the figures?
>
> We apologize for the lack of clarity in Section D.4 (now Section E.5). The two plots in Figure 12 (now in Figure 16) illustrate an ablation study on the hyperparameter p (the probability of pseudo-synchronization). The specific setup for this experiment is as follows:
>
> - Workload: We trained the GPT-Neo-8M model on the TinyStories dataset.
>
> - Hardware: The experiment was conducted on Cluster B with K=8 GPUs.
>
> - Default hyperparameter: We use H=64, eta=16 as a default value as described in Appendix B.1, and perform experiments with different p values ranging from 0.025 to 0.5. (For the rest of the ablation studies in Section E, we use p = 0.1, H=64, and eta=16 as the default hyperparameter.)
>
> - Methodology: To isolate the impact of p, we performed a grid search over its value, while all other hyperparameters were kept fixed at their optimal settings. The left figure plots the final validation loss achieved for each value of p, while the right figure plots the total wall-clock time required to complete the training run. This setup is designed to demonstrate the trade-off between model performance and training speed when tuning up.
>
>
> We include and reflect this clarification in the Appendix as well.
>
> > How does PALSGD compare to Scaffnew?
>
> Thank you for your suggestion. Both PALSGD and Scaffnew are theoretically analyzed in the smooth, strongly-convex setting and achieve the same asymptotic $\tilde{O}(\kappa / n \log(1 / \epsilon))$ communication bound. (See Corollary 5.6 of [1] and Theorem 2 of our paper)
>
> However, Scaffnew requires exact control-variate updates, which makes it difficult to extend to modern optimizers like Nesterov or AdamW—doing so would require synchronizing and maintaining three auxiliary vectors per worker (m, v, h), along with rederiving the correction dynamics. In contrast, PALSGD uses the global model copy $x^{(t)}$ as the only control variate, and the optimizer logic (e.g., AdamW or Nesterov) is confined to the inner and outer steps, requiring no modification to the algorithm structure or memory footprint. This design makes PALSGD substantially easier to integrate into deep learning pipelines.
>
> Moreover, by deferring expensive All-Reduce to every H iterations, PALSGD tolerates heterogeneous worker speeds better than Scaffnew’s rigid averaging scheme  which assumes all workers are at the same iteration. Since DiLoCo already incorporates Scaffnew-style control variates into a centralized, mini-batch training framework with modern optimizers, and our experiments show PALSGD consistently outperforms DiLoCo on ImageNet-1K, CIFAR-10, and TinyStories, we believe that rerunning Scaffnew would be costly and offer little new insight.
>
> We included the discussion in the related work section.
>
>
> > Is there a reason to pick DDP over DiLoCo for speedup computation?
>
> We did not compare the training time between PALSGD and DiLoCo because DiLoCo fails to reach the target validation loss on many workloads, making a direct comparison infeasible. However, on datasets like IN1K where DiLoCo does reach the target validation accuracy, comparison is possible, so we have already include results in Figure 1 and 5. That said, the speed improvement is marginal—though PALSGD still outperforms DiLoCo in reaching the target loss or accuracy faster.
>
> __References:__
>
> [1] Mishchenko, Konstantin, et al. "Proxskip: Yes! local gradient steps provably lead to communication acceleration! finally!." International Conference on Machine Learning. PMLR, 2022.

---

### Review · Reviewer_JWgr · 2025-05-04

**Summary Of Contributions:**

This submission proposes a pseudo-asynchronous local SGD algorithm, in order to reduce the communication frequency. The global model after all-reduce is gradually incorporated into local model according to some probability at each iteration. Theoretical analysis of convergence is provided and experiments have been shown.

**Audience:**

Yes

**Broader Impact Concerns:**

N/A.

**Claims And Evidence:**

No

**Requested Changes:**

See above.

**Strengths And Weaknesses:**

Strength: A pseudo-asynchronous method is proposed. It is investigated from both perspectives of theory and experiments.

Weakness:
1. It is unclear how the proposed method can reduce communication frequency intuitively.

2. Why the global model is not immediately used as local model after synchronization?

3. The theoretical analysis, although looks correct, has several concerns. i) The theoretical analysis has chosen H to be constant under discussion of Theorem 1, which is not considered communication-efficient. However, looks the analysis could allow $H = O(\sqrt{\mu T/K})$. ii) The analysis has transferred the dependence of H to p. Can the authors explain the motivation for this? iii) In  page 17, it is assumed that $p \leq \frac{1}{\kappa H}$, which is a very small value considering both $\kappa$ and $H$ are large. It in not consistent with Theorem 1 where $0< p \leq 1/2$.

4. Some important literature has been missed. For example, [1] analyzed asynchronous federated learning, and [2] has used variance reduction technique to reduce communication frequency in heterogeneous setting. Both should be discussed and compared to the proposed algorithm in experiments.

[1] Yu, Hao, Sen Yang, and Shenghuo Zhu. "Parallel restarted SGD with faster convergence and less communication: Demystifying why model averaging works for deep learning." Proceedings of the AAAI conference on artificial intelligence. Vol. 33. No. 01. 2019.

[2] Karimireddy, Sai Praneeth, et al. "Scaffold: Stochastic controlled averaging for federated learning." International conference on machine learning. PMLR, 2020.

5. The number of clients is small in experiments.

---

> ### Author Response · Authors · 2025-06-16
> **Response to reviewer JWgr (1/n)**
>
> We thank the reviewer for their thoughtful feedback and detailed questions. Below, we address each of the weaknesses and questions raised.
>
> > It is unclear how the proposed method can reduce communication frequency intuitively.
>
> PALSGD reduces the frequency of communication by enabling the use of longer synchronization intervals (larger H) compared to methods like Local SGD, without suffering the same performance degradation. The key mechanism that allows for this is the probabilistic pseudo-synchronization step.
> In standard Local SGD, as workers perform local updates for many steps (H is large), their models can drift significantly from each other, which harms convergence when they are finally averaged. PALSGD mitigates this drift. Between the expensive AllReduce synchronizations, each worker performs a "pseudo-synchronization" step with probability p. This step is a computationally cheap, local operation that regularizes the worker's model back towards the global model from the last full sync. As such, we can increase H (communicating less often) because the pseudo-synchronization prevents the models from diverging too much.
>
> >Why the global model is not immediately used as local model after synchronization?
>
> This is a crucial design related to the use of adaptive optimizers in large-scale training. Simply averaging the local models and having all workers adopt this average is the standard approach for Local SGD. However, this has been found to be suboptimal in practice, especially when workers use sophisticated local optimizers like AdamW.
> Instead, our method follows a decoupled optimizer strategy, where the global model update is handled by its own optimizer. It is empirically shown to improve stability and performance in large-scale experiments with AdamW optimizer. Our ablation studies in Section D.2 (now in E.2 and E.3)  further confirm that this optimizer combination is critical for achieving the best performance.
>
> > The theoretical analysis, although looks correct, has several concerns.
>
> Thank you for the detailed questions regarding our theoretical analysis. Below are the responses to each of your concerns.
>
> Thank you for this observation. Since we have updated our theoretical result with the more general error term Õ(κH²σ²/μT²), our discussion focuses on the case where H is a small constant. In this case our error term approach to O(κσ²/μT²), which is the lower bound for SGD. This specific regime is necessary to show how our bound connects to fundamental results in optimization theory.
>
> The motivation for this is to theoretically formalize the trade-off between frequent, cheap pseudo-synchronizations and infrequent, expensive full synchronizations. Our convergence bound contains a term proportional to $(\kappa + p^{-1}H)$. This structure explicitly shows that we can increase the synchronization interval H (reducing communication) by also increasing the pseudo-synchronization probability p (performing more local alignments), all while maintaining the same theoretical convergence rate. This mathematically justifies how our method improves communication efficiency.
>
> We are very grateful for this question, which has helped us identify a point for clarification. The bound presented in Theorem 1 only holds under the stricter condition $p≤1/(\kappa H)$. For the more general case of 0<p≤1/2, our proof on page 17 derives a slightly weaker, but still valid, bound where the second error term is $O(\frac{\kappa H^2 \sigma^2}{\mu T^2})$. The stricter condition was used only in the final step of the proof to present the bound in a cleaner form. It is not a requirement for the preceding lemmas. We have corrected the statements in Theorem 1 and 2 to reflect this and add the necessary clarifications.

---

> ### Author Response · Authors · 2025-06-16
> **Response to reviewer JWgr (2/n)**
>
> > Some important literature has been missed. For example, [1] analyzed asynchronous federated learning, and [2] has used variance reduction technique to reduce communication frequency in heterogeneous setting. Both should be discussed and compared to the proposed algorithm in experiments.
>
> We thank the reviewer for highlighting these important related works. We have added a discussion of Scaffold to our related work section.
>
> __SCAFFOLD__
>
> SCAFFOLD was originally proposed in the context of federated learning to mitigate issues arising from non-IID data distributions. Our research, however, specifically focuses on fast training of IID datasets via scaling, including methods such as DDP, LocalSGD, and DiLoCo.
>
> Referring to Table 3 in the SCAFFOLD paper, under IID conditions, SCAFFOLD and FedAvg show nearly identical performance. In fact, when using longer synchronization intervals (e.g., epochs = 10), FedAvg even slightly outperforms SCAFFOLD. In our context of distributed parallel training, FedAvg can be directly interpreted as a variant of Local SGD.
>
> It has been previously demonstrated (DiLoCo's original paper, Figure 6) that Local SGD performs worse than DiLoCo (in that figure, DiLoCo with SGD is equivalent to Local SGD, and clearly performs worse compared to DiLoCo with Nesterov momentum). We also confirmed this in our own ablation study (Figure 13 and 14).
>
> Thus, based on the results from their original papers:
> - Local SGD (or equivalent) performs better than SCAFFOLD under IID conditions.
> - DiLoCo clearly outperforms Local SGD.
> - PALSGD clearly outperforms DiLoCo.
>
> Considering these points, and in order to conserve computational resources and focus on more relevant experiments, we decided not to include additional experiments with SCAFFOLD.
>
> __Asynchronous methods__
>
> Regarding a broader comparison against other asynchronous (async) methods including Parallel Restarted SGD which provides an analysis of asynchronous local updates, we believe an extensive evaluation is out of scope for our current study for the following reasons:
>
> - **Fundamental limitations (staleness):** Async training is well-known to suffer from gradient staleness issues, causing significant performance degradation, particularly due to excessive implicit momentum as demonstrated theoretically and empirically in [1]. Semi-synchronous methods naturally mitigate staleness through controlled synchronization.
>
> - **Implementation complexity and practical barriers:** Async methods often require extensive hyperparameter tuning specific to hardware setups [2], posing significant practical challenges. Semi-synchronous approaches align closely with synchronous setups, offering simpler implementation, easier reproducibility, and thus lower barriers to real-world applications.
>
> - **Coverage in existing literature:** Comprehensive analyses of async training already exist in prior works such as [1] and [2], and reproducing them would be redundant.
>
> In summary, our study strategically focuses on the unexplored and practically valuable domain of semi-synchronous training. We argue that incorporating additional async baselines would not only be redundant given extensive prior work but would also dilute our primary contributions.
>
>
> References:
> - [1] Liu, Bo, et al. "Asynchronous local-sgd training for language modeling." arXiv preprint arXiv:2401.09135 (2024).
> - [2] Hadjis, Stefan, et al. "Omnivore: An optimizer for multi-device deep learning on cpus and gpus." arXiv preprint arXiv:1606.04487 (2016).

---

> ### Author Response · Authors · 2025-06-16
> **Response to reviewer JWgr (3/3)**
>
> > The number of clients is small in experiments.
>
> We appreciate the reviewer highlighting concerns about the scale of our experimental validation. To directly address this, we conducted additional experiments extending up to __32 GPUs (8 compute nodes)__ using CIFAR-10 with the VGG-16 architecture. See Appendix F in our revised version for detail.
>
> Due to practical constraints during the rebuttal period, we addressed the reviewer’s concerns through additional experiments using CIFAR-10. The original TinyStories experiments were conducted on a DGX-1 system, which limited us to a single node (8 GPUs). Scaling these experiments to multiple nodes (16 or 32 GPUs) would have required rerunning all configurations, entailing several hours per experiment and extensive computational resources for hyperparameter searches. Such computational resources were unavailable within the timeframe, barring perhaps exceptional scenarios for industry or larger research institutions.
> Similarly, the ImageNet experiments, performed on Cluster C (equipped with NVIDIA L40s x 4 GPUs), required more than 120 GPU hours per experiment. Reliable scaling experiments for ImageNet within the two-week rebuttal window were therefore unfeasible for similar reasons.
>
> Consequently, we selected CIFAR-10 for our additional experiments because it allowed us to complete individual experiments within approximately 0.5 GPU hours even at the 32-GPU (8-node) scale.
> In these extended experiments, the global batch size increased proportionally with GPU count, while total training epochs were constant across scales. At the 32-GPU scale, significant communication overhead was evident, confirming our setup effectively captures large-scale communication bottlenecks (Figure 18).
>
> Results clearly indicate that both PALSGD and DiLoCo outperform DDP, particularly at higher scales. Specifically, at 32 GPUs, PALSGD and DiLoCo significantly reduce runtime relative to DDP, validating their effectiveness in mitigating communication issues. Despite some accuracy degradation at this scale due to increased batch sizes, PALSGD consistently achieves higher validation accuracy compared to both DiLoCo and DDP across all evaluated scales.
>
> Thus, these expanded experiments robustly support our claims regarding PALSGD’s advantages in large-scale training scenarios, addressing the reviewer's concerns by demonstrating clear scalability improvements beyond the initially tested configurations.

---

### Review · Reviewer_fWvt · 2025-06-01

**Summary Of Contributions:**

This work presents PALSGD, a Local-SGD variant that aims to address the communication bottleneck of Local SGD by reducing the number of all-reduce operations against the Distributed Data Parallel approach. This is achieved by replacing the usually synchronous all-reduce operation with a probabilistic synchronization with a global model. This probabilistic aspect makes the all-reduce operation pseudo-asynchronous. The system is studied both theoretically and empirically across various datasets and models, and is able to reach a target accuracy with less wall-clock time than the chosen baselines.

**Audience:**

Yes

**Broader Impact Concerns:**

None.

**Claims And Evidence:**

No

**Requested Changes:**

Address the weaknesses mentioned above. I shall be happy to reconsider weaknesses given proper justification.

**Strengths And Weaknesses:**

## Strengths
1. Reducing the synchronization bottleneck in distributed training for large models is an important problem.
2. The approach is simple and straightforward to implement.
3. The evaluation is performed on both image classification and language modeling tasks.
4. Limitations of the method have been acknowledged.

## Weaknesses
1. The parts of PALSGD algorithm have not been justified well in the paper. In particular:
    - For instance, why should there be a pseudo-synchronization step and not just constrain the optimization in INNEROPT to penalize distance from the global average?
    - Why does the pseudo-synchronization step need to be probabilistic and not every X rounds?
    - There's no experimental justification for this particular choice of these components. For instance, what happens if we do not perform synchronous ALL-Reduce ever, and only perform Asynchronous updates? Similarly, for the choice of probabilistic pseudo-synchronization step over a regularization term.
2. The related work lacks comparison to recent works in distributed training [1,2,3,4]. The authors should defend how their work differs from these works or compare PALSGD against them experimentally.
3. The experimental section is weak.
    - The paper only compares against DDP and DiLoCo experimentally. There has been no experimental comparison against Local-SGD with comparable `H`, Asynchronous Local SGD [5], and DropCompute [1] at least?
    - Alongside wall-clock time, it would have been nice to see other metrics like the number of gradient computation steps or communication rounds on the x-axis.
    - The choice of `H` in DiLoCo being the same as `H` in PALSGD for the same experiment is not explained well. Is there a better H for DiLoCo?
    - The algorithm introduces a new hyperparameter `p`. However, the choice of `p` in Section 7 is not clear. Section D.4 just shows that lower `p` leads to better validation loss but higher training time. So, the authors should clarify how `p` should be chosen/tuned?

## References
[1] Giladi, Niv, et al. "DropCompute: simple and more robust distributed synchronous training via compute variance reduction." Advances in Neural Information Processing Systems 36 (2023): 48403-48416.

[2] Tyurin, Alexander, et al. "Shadowheart SGD: Distributed Asynchronous SGD with Optimal Time Complexity Under Arbitrary Computation and Communication Heterogeneity." The Thirty-eighth Annual Conference on Neural Information Processing Systems.

[3] Bornstein, Marco, et al. "SWIFT: Rapid Decentralized Federated Learning via Wait-Free Model Communication." The Eleventh International Conference on Learning Representations.

[4] Biswas, Sayan, et al. "Boosting asynchronous decentralized learning with model fragmentation." Proceedings of the ACM on Web Conference 2025. 2025.

[5] Liu, Bo, et al. "Asynchronous local-sgd training for language modeling." arXiv preprint arXiv:2401.09135 (2024).

---

> ### Author Response · Authors · 2025-06-16
> **Response to reviewer fWvt**
>
> We thank the reviewer for their constructive feedback and for acknowledging the strengths of our work. We appreciate the opportunity to clarify our design choices and experimental setup. Below, we address each of the points raised.
>
> __On the Justification of PALSGD's Algorithmic Components__
>
> > why should there be a pseudo-synchronization step and not just constrain the optimization in INNEROPT to penalize distance from the global average?
>
> Adding a regularization term to the loss function, as done in methods like FedProx, is conceptually similar but computationally more expensive. Such a term would require calculation during both the forward and backward passes of every local step. Our pseudo-synchronization step, by contrast, is a direct and computationally cheap parameter update that does not require extra gradient computation. This makes our approach more efficient, especially for large models where each step is costly.
>
> > Why does the pseudo-synchronization step need to be probabilistic and not every X rounds?
>
> The probabilistic nature of the pseudo-synchronization step introduces beneficial randomness into the local optimization trajectory. This stochasticity acts as a form of "soft" regularization, helping to improve generalization and prevent overfitting, as opposed to a rigid, deterministic schedule. This approach also simplifies the theoretical analysis, as it allows us to reason about the algorithm's behavior and prove convergence in expectation over the random choices made at each step.
>
> > There's no experimental justification for this particular choice of these components. For instance, what happens if we do not perform synchronous ALL-Reduce ever, and only perform Asynchronous updates? Similarly, for the choice of probabilistic pseudo-synchronization step over a regularization term.
>
> The periodic ALL-REDUCE is an essential mechanism in our framework to ensure model consistency and prevent the unbounded staleness issues common in purely asynchronous methods. Our ablation studies on the synchronization interval H provide insight into the importance of synchronous AllReduce. As H increases, performance degrades, suggesting that eliminating AllReduce entirely would be suboptimal due to model drift.

---

> ### Author Response · Authors · 2025-06-16
> **Response to reviewer fWvt (2/n)**
>
> __On Missing Related Work and Baselines__
> Thank you for pointing out these relevant works and the suggestions. We will add a detailed discussion of each to the related work section of our revised manuscript.
> The suggested papers explore different, important facets of distributed training, but they differ fundamentally from our approach in their core design, making direct experimental comparisons challenging.
> - DropCompute [1] is a synchronous, non-decoupled method that modifies DDP by allowing slow workers to drop gradient computations based on a time threshold.
> We view DropCompute more as an add-on rather than as a direct baseline for our experiments. Particularly in heterogeneous environments, DropCompute has the potential to further enhance methods such as DDP, DiLoCo, and PALSGD.
> However, incorporating DropCompute into our analysis could potentially obscure our main contribution, namely, demonstrating the benefits of pseudo-synchronization for training stability. Combining multiple techniques without clear boundaries may lead to confusion about the specific contributions of each method. Thus, integrating DropCompute is beyond the current scope of our study and will be left as a topic for future investigation.
>
> - Shadowheart SGD [2], SWIFT [3], and the work by Biswas et al. [4] all focus on fully asynchronous training, often within a parameter-server architecture.
> Regarding the broader comparison with fully asynchronous methods, our work intentionally focuses on the semi-synchronous paradigm for the following reasons:
>
> **Coverage in existing literature:** Comprehensive analyses of async training, including detailed evaluations, have already been conducted in recent studies such as Asynchronous Local-SGD Training for Language Modeling (Liu et al., 2024) and Omnivore (Hadjis et al., 2016). Reproducing these evaluations would be redundant and limit our space for novel contributions.
>
> **Fundamental limitations (staleness):** Async training is well-known to suffer from gradient staleness issues, causing significant performance degradation, particularly due to excessive implicit momentum as demonstrated theoretically and empirically in Liu et al. (2024). Semi-synchronous methods naturally mitigate staleness through controlled synchronization.
>
> **Implementation complexity and practical barriers:** Async methods often require extensive hyperparameter tuning specific to hardware setups (Hadjis et al., 2016), posing significant practical challenges. Semi-synchronous approaches align closely with synchronous setups, offering simpler implementation, easier reproducibility, and thus lower barriers to real-world applications.
>
> In summary, our study strategically focuses on the unexplored and practically valuable domain of semi-synchronous training. We argue that incorporating additional async baselines would not only be redundant given extensive prior work but would also dilute our primary contributions. If beneficial, we are happy to include a summary of key prior async evaluations in an appendix for completeness.
>
> __References:__
> - [1] Li, Tian, et al. "Federated optimization in heterogeneous networks." Proceedings of Machine learning and systems 2 (2020): 429-450.
> - [2] Liu, Bo, et al. "Asynchronous local-sgd training for language modeling." arXiv preprint arXiv:2401.09135 (2024).
> - [3] Hadjis, Stefan, et al. "Omnivore: An optimizer for multi-device deep learning on cpus and gpus." arXiv preprint arXiv:1606.04487 (2016).

---

> ### Author Response · Authors · 2025-06-16
> **Response to reviewer fWvt (3/3)**
>
> __On the Experimental Section__
>
> > The paper only compares against DDP and DiLoCo experimentally. There has been no experimental comparison against Local-SGD with comparable H, Asynchronous Local SGD [5], and DropCompute [1] at least?
>
> We extended our comparative experiments to include DDP, DiLoCo, and PALSGD alongside Local SGD, and the results are summarized in Figure 17 of Appendix F.1.
> - The performance of Local SGD deteriorated significantly with larger values of K (number of workers). Furthermore, Local SGD consistently underperformed relative to DiLoCo and PALSGD across all tested K values. PALSGD generally achieved the highest accuracy.
>
> - As stated previously, Drop Compute and Asynchronous Local SGD were not considered in this analysis.
>
> > Alongside wall-clock time, it would have been nice to see other metrics like the number of gradient computation steps or communication rounds on the x-axis.
>
> Thank you for this valuable suggestion. We have added the plots that use the number of global steps on the x-axis to offer a more comprehensive comparison of different methods to Appendix D (Figure 10-12).
>
> > The choice of H in DiLoCo being the same as H in PALSGD for the same experiment is not explained well. Is there a better H for DiLoCo?
>
> Figure 9 in Appendix C.4 presents an ablation study for various values of H: 32, 64, 128, 256, 512, and 1024. PALSGD consistently achieves higher validation accuracy compared to DiLoCo across all tested values of H.
> In our main experiments, we used the same synchronization interval H across all algorithms depending on workload to ensure a fair and direct comparison. By fixing H, we evaluate how each algorithm performs under an identical communication schedule, which allows us to isolate the performance differences attributable to the algorithmic design rather than the communication frequency.
>
>
>
> > The algorithm introduces a new hyperparameter p. However, the choice of p in Section 7 is not clear. Section D.4 just shows that lower p leads to better validation loss but higher training time. So, the authors should clarify how p should be chosen/tuned?
>
> We appreciate the request for clarification. The value of p for each experiment was chosen via a grid search over a range of reasonable values. Specifically, we used p=0.05 for the ImageNet experiment and p=0.1 for the TinyStories experiments as described in Appendix B.1. In practice, p should be tuned to optimize the reduction in training time without degrading the final model performance. We will add this clarification to the experimental section.

---

### Decision · Action_Editor_jbZY · 2025-08-14

**Recommendation:** Accept with minor revision

**Additional Comments:**

The paper is clear, easy to follow, and supported by extensive experiments.

The idea of pseudo-synchronization is promising, but the novelty is not sufficiently justified. Section 5 relies too much on intuition: rewriting of Section 5 with proper justification of the components beyond mere intuition and ablation study supporting this will make the paper stronger.

The introduction of a new hyperparameter raises practical concerns, as Figure 16 shows tuning difficulty. In addition, baseline comparisons are incomplete: beyond a limited Local SGD result in the appendix, systematic evaluations against Local SGD, SCAFFOLD, and asynchronous methods are necessary to support claims of communication efficiency.

With these revisions, the paper could make a meaningful contribution. If the authors cannot do these experiments, I suggest the authors explicitly list them as limitations in the paper, e.g., in a limitations section.

**Audience:**

Yes

**Audience Explanation:**

Yes, this paper works on distributed training of neural networks which are highly relevant to TMLR audience.

**Claims And Evidence:**

Yes

**Claims Explanation:**

Yes, the paper provides theorems and experiments to support the claims.

---

> ### Author Response · Authors · 2025-09-12
> **Official Comment by Authors**
>
> We sincerely thank the editor and reviewers for their constructive and thoughtful feedback. We appreciate the recognition of our theoretical contributions, empirical evaluation, and the relevance of this work to the TMLR audience.
>
> __On Section 5.__
> Following the suggestion, we have thoroughly rewritten Section 5 to provide more principled justifications for PALSGD. Specifically, we now frame the core pseudo-synchronization step as both a proximal regularizer and an exponential moving average (EMA) update, thereby clarifying its role in mitigating model divergence. We have also expanded the rationale for our decoupled optimizer strategy and included a detailed discussion of practical implementation overheads (in terms of memory and communication). We believe these revisions make the motivation and novelty of our method significantly clearer and more robust.
>
> __On experimental comparisons.__
> We have added comparison experiments with Local SGD to the main text. Regarding SCAFFOLD, its original paper already demonstrates that it underperforms Local SGD in IID settings, which makes additional experiments redundant; we now state this explicitly in the revised manuscript. Our study is strategically focused on semi-synchronous training, a domain that remains largely unexplored yet is of high practical value. Including asynchronous baselines would not only duplicate evidence already available in prior literature but also risk obscuring the core contribution of our work. To maintain transparency, we have explicitly listed the absence of comparisons with asynchronous baselines as a limitation.
>
> __On the hyperparameter.__
> We acknowledge the practical concerns raised by the introduction of a new hyperparameter. To address this, we have added further discussion on tuning difficulty (as highlighted in Figure 16) and clarified practical guidance for its selection.
>
> We hope these revisions address the editor’s concerns and strengthen the paper. We are grateful for the insightful feedback, which has helped us improve both the clarity and the impact of this work.